# *In vivo* gene expression profile of *Haemophilus influenzae* during human pneumonia

Linnea Polland,[1,2] Hanna Rydén,[2,3] Yi Su,[1] Magnus Paulsson[1,2]

**ABSTRACT**   *Haemophilus influenzae* is a major cause of community-acquired pneumonia. While studied extensively in various laboratory models, less is known about the cell function while inside the human lung. We present the first analysis of the global gene expression of *H. influenzae* while the bacteria are in the lung during pneumonia (*in vivo* conditions) and contrast it with bacterial isolates that have been cultured under standard laboratory conditions (*in vitro* conditions). Patients with pneumonia were recruited from emergency departments and intensive care units during 2018–2020 ($n = 102$). Lower respiratory samples were collected for bacterial culture and RNA extraction. Patient samples with *H. influenzae* ($n = 8$) and colonies from bacterial cultures ($n = 6$) underwent RNA sequencing. The reads were then pseudo-aligned to core and pan genomes created from 15 reference strains. While bacteria cultured *in vitro* clustered tightly by principal component analysis of core genome ($n = 1067$) gene expression, bacteria in the patient samples had more diverse transcriptomic signatures and did not group with their lab-cultured counterparts. In total, 328 core genes were significantly differentially expressed between *in vitro* and *in vivo* conditions. The most highly upregulated genes *in vivo* included *tbpA* and *fbpA*, which are involved in the acquisition of iron from transferrin, and the stress response gene *msrAB*. The biosynthesis of nucleotides/purines and molybdopterin-scavenging processes were also significantly enriched *in vivo*. In contrast, major metabolic pathways and iron-sequestering genes were downregulated under this condition. In conclusion, extensive transcriptomic differences were found between bacteria while in the human lung and bacteria that were cultured *in vitro*.

**IMPORTANCE**   The human-specific pathogen *Haemophilus influenzae* is generally not well suited for studying in animal models, and most laboratory models are unlikely to approximate the diverse environments encountered by bacteria in the human airways accurately. Thus, we have examined the global gene expression of *H. influenzae* during pneumonia. Extensive differences in the global gene expression profiles were found in *H. influenzae* while in the human lung compared to bacteria that were grown in the laboratory. In contrast, the gene expression profiles of isolates collected from different patients were found to cluster together when grown under the same laboratory conditions. Interesting observations were made of how *H. influenzae* acquires and uses iron and molybdate, endures oxidative stress, and regulates central metabolism while in the lung. Our results indicate important processes during infection and can guide future research on genes and pathways that are relevant in the pathogenesis of *H. influenzae* pneumonia.

**KEYWORDS**   *Haemophilus influenzae*, pneumonia, gene expression, mRNA, metabolic modeling

*H*aemophilus influenzae is the second most common infectious bacterial agent responsible for community-acquired pneumonia and the most frequent bacterial

Address correspondence to Magnus Paulsson, magnus.paulsson@med.lu.se.

The authors declare no conflict of interest.

See the funding table on p. 19.

cause of exacerbations of chronic obstructive pulmonary disease (COPD) (1, 2). The severity of pneumonia caused by *H. influenzae* ranges from mild to very severe, and symptoms typically involve acute onset of cough and fever.

*H. influenzae* has evolved to be a pathogenic bacterial species that survives exclusively in the human respiratory tract. It carries a relatively small genome of about 1,800,000 bp that is well defined, and several annotated reference genomes exist, including the strain Rd KW20 (3). On average, 47% of *H. influenzae* genes are shared between all strains and belong to the core genome (4). Despite the genetic variations, its small genome necessitates low genetic redundancy, making *H. influenzae* an attractive model organism for human-pathogen interaction studies. Such studies have been performed mainly using laboratory or animal models. However, animal models are complicated by the restricted niche of *H. influenzae*; laboratory models for studies of bacterial physiology or host-pathogen interactions inevitably introduce simplifications and experimental biases. For this reason, they only partly reflect the complexity of the environment *in vivo*. Transcriptomic studies of other bacterial species have revealed a significant difference in gene expression patterns between bacterial cells cultured in any *in vitro* model and those that grow *in vivo* (5). This discrepancy is attributed to variations in physiological parameters *in vivo*, including altered temperature and oxygen availability, as well as limitations in nutrient availability and bacterial stresses caused by host-pathogen interactions and administered drugs.

Previously, clinical *H. influenzae* strains have been co-cultured with human cell lines in dual RNA-seq experiments focusing on host-pathogen interactions—a model that lacks the complexity of the environment in human airways (6, 7). Other studies have described the *in vivo* gene expression of *H. influenzae* using various animal models (8, 9). *Haemophilus influenzae* is an obligate human pathogen, and its transcriptome during infection of a human host is largely unknown. We hypothesize that there is a significant discrepancy in transcriptomic signatures between *H. influenzae* cells that are cultured in a laboratory environment and those that grow in the human lung during acute pneumonia and that this difference can be used to find biological processes that are important for the pathogenesis and persistence of *H. influenzae in vivo*. In the present study, we aimed to elucidate this difference by comparing the gene expression profile of *H. influenzae* that grew in the human lung versus bacteria that were cultured under typical *in vitro* conditions.

## MATERIALS AND METHODS

### Study participants and sample collection

Study participants were recruited among patients who were admitted to the emergency departments of Skåne University Hospital (Lund and Malmö, Sweden) between June 2019 and May 2020. Inclusion criteria were age 18 years or older and two of the following signs: productive cough, dyspnea, and/or fever (≥38.5℃). In addition, participants were recruited among patients admitted to the intensive care unit (ICU) of Skåne University Hospital (Malmö, Sweden) between July 2018 and March 2020 (Fig. 1). The inclusion criteria for these latter subjects were age 18 years or older and clinician's suspicion of pneumonia based on the decision to collect respiratory samples for microbiological analysis. Sputum or bronchoscopy-guided bronchial wash samples (using 2 × 10 mL sterile 0.9% saline) were collected. A portion of the samples was sent for bacterial culture; RNAlater Solution (Thermo Fisher Scientific, Waltham, MA, USA) was added to the remaining portion of the samples within 1 minute of collection to preserve RNA integrity. Samples with RNAlater were stored at −20℃. Study data were collected from the hospital records using an electronic case report form (REDCap), hosted at Lund University (10). The present project was approved by the Swedish Ethical Review Authority (Dnr 2019-01012). All patients were given oral and written information about the study and signed a consent form.

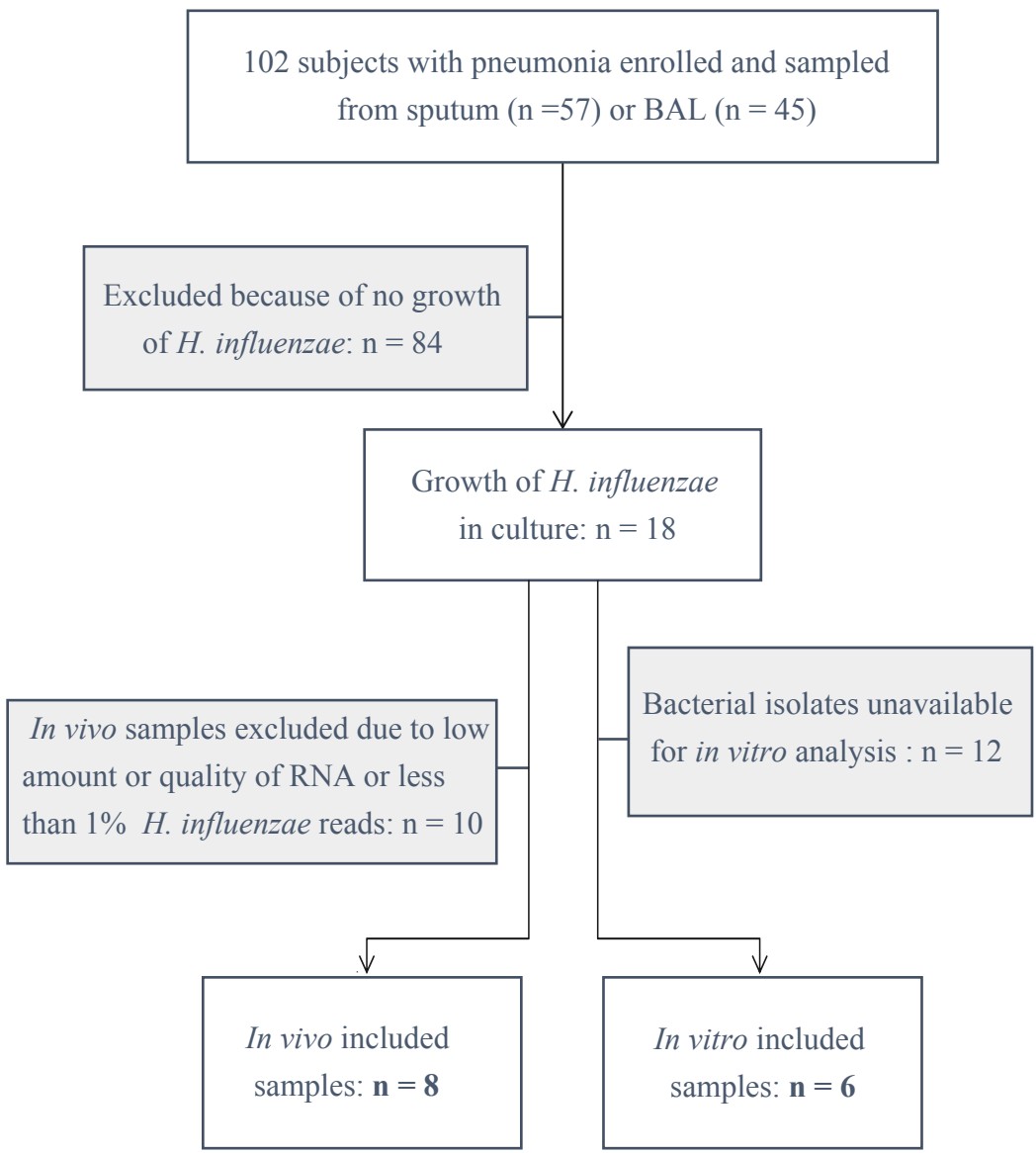

**FIG 1** Flowchart of study subject and sample inclusion. Of the 18 (out of 102) collected samples containing culturable *H. influenzae,* six were saved by the clinical microbiology department and available for *in vitro* cultures. Of the 18 corresponding respiratory tract *in vivo* samples stored in RNA-preserving buffer, 8 contained sufficient amounts of quality RNA for sequencing and, post-sequencing, at least 1% *H. influenzae* transcripts and could be used for further analysis.

## Bacterial strains and culture conditions

The patient samples underwent standard bacterial culturing in the Clinical Microbiology division of the Laboratory Medicine unit of Region Skåne (Lund, Sweden). Bacterial species were identified based on phenotype and MALDI-TOF (matrix-assisted laser desorption/ionization—time-of-flight). According to the study protocol, *H. influenzae* isolates were placed in glycerol stock and frozen at −80℃ for subsequent analysis. For comparison, a reference strain, non-typeable *H. influenzae* 3655, was kindly provided by Prof. Riesbeck (Lund University, Sweden) and included in the analysis.

For the *in vitro* gene expression analysis, two distinct colonies of *H. influenzae* from the primary culture plates were selected and each one was inoculated to an initial $OD_{600}$ of 0.05 in brain heart infusion (BHI) broth that contained 10 µg/mL nicotinamide adenine

dinucleotide and 10 µg/mL hemin. The isolates were cultured at 37°C in 5% $CO_2$ with shaking at 200 rpm, stopped in the exponential phase after 240 minutes ($OD_{600}$ 0.5–0.6), and centrifuged for 2 minutes at 10,000 × $g$ to obtain a pellet. The supernatant was removed, and RNAlater was added to the samples.

## DNA extraction, sequencing, and base-calling

Genomic DNA was extracted from bacterial colonies that were harvested from chocolate blood agar plates that had been cultured for 16 h at 37°C and 5% $CO_2$ using the MagAttract HMW DNA kit (Qiagen, Hilden, Germany) following the manufacturer's instructions. Libraries for DNA sequencing were prepared with the RBK-004 Rapid Barcoding Kit (Nanopore, Oxford, United Kingdom). DNA sequencing was performed on a Nanopore Mk1C with an R9.4.1 Flow Cell for 24 h. The resulting read files were base-called, and genomes were assembled and confirmed as *H. influenzae* using Epi2me-labs wf-bacterial-genomes/0.2.12 with the standard setting (Nanopore). The multilocus sequence type (MLST) of each assembled genome was determined according to PubMLST (https://pubmlst.org) and assemblies were predicted to an *H. influenzae* serotype based on the cap locus using hicap/1.0.3 (11).

## RNA extraction from clinical samples and cultured bacteria

Bacterial suspensions and clinical samples in RNAlater were processed at room temperature. Samples were centrifuged at 7,000 × $g$ for 30 minutes, and the supernatant was removed, except for 1 mL that was left with the pellet. Next, 1 mL TE buffer (10 mM Tris, 1 mM EDTA; Thermo Fisher Scientific) with 2 mM dithiothreitol (DTT; Thermo Fisher Scientific) was added, and the suspension was centrifuged at 1,000 × $g$ for 15 minutes. The supernatant was discarded, and the pellet was resuspended in 563 µL TE buffer with 2 mM DTT, lysozyme (1 mg/mL; Thermo Scientific, Rockford, IL, USA), and lysostaphin (0.17 mg/mL; Sigma-Aldrich, St. Louis, MO, USA). The resuspension was then transferred to a tube containing 1.4 mm ceramic beads, 0.1 mm silica beads, and a 4-mm glass bead (MP Biomedicals, Santa Ana, CA, USA); homogenized by bead beating at 4,350 rpm (BeadBug 6; Benchmark scientific, Sayreville, NJ, USA) for 30 s; and incubated for enzymatic lysis at 37°C for 30 minutes. Three milliliters of Trizol LS (Thermo Fisher Scientific) was added, and the sample was mixed, split into bead beating tubes with Matrix E (MP Biomedicals), and homogenized at 4,350 rpm for 3 × 90 s. Next, 0.27 mL chloroform was added, and RNA was extracted from the aqueous phase by isopropanol precipitation. The RNA pellet was washed three times with 75% ethanol, air-dried, and resuspended in 50 µL RNase-free water. RNA concentrations were quantified on a NanoDrop spectrophotometer (Thermo Fisher Scientific) and using the QuantIT RNA HS assay kit on a Qubit 4 fluorometer (Thermo Fisher Scientific). RNA integrity was evaluated on a 2100 Bioanalyzer capillary gel electrophoresis instrument (Agilent, Santa Clara, CA, USA), and RIN values were determined. Samples with highly degraded RNA (RIN < 2) were not processed further.

## rRNA depletion, library preparation, and sequencing

Bacterial ribosomal RNA (rRNA) was depleted from 100 ng of total RNA with QIAseq Fastselect 5S/16S/23S (Qiagen), without a fragmentation step. Human rRNA depletion and library preparation were performed with the SMARTer Stranded Total RNA-Seq Kit v2—Pico Input Mammalian Components (Takara, Kusatsu, Japan) kit according to the manufacturer's instructions. Library size and quality were assessed with Agilent D5000 screentape and reagents on a 4200 TapeStation system (Agilent). Samples without detectable libraries were excluded from further analysis. An initial "scout" sequencing run was paired-end (resulting in two files, designated R1 and R2), and subsequent sequencing was single-ended (resulting in only 1 R1 file). The libraries, at a concentration of 0.65 nM, was loaded onto NovaSeq 6000 S1 (100 cycles) or NovaSeq 6000 SP (200 cycles) flow cells and sequenced on a NovaSeq 6000 system (Illumina, San Diego, CA, USA) with 1% PhiX control by the Center for Translational Genomics, Lund University.

## RNA-sequencing file processing and taxonomic assignment

Initial computations were performed on the Uppsala Multidisciplinary Center for Advanced Computational Science (UPPMAX) Bianca server, a high-performance and General Data Protection Regulation (GDPR) - compliant computing cluster. Demultiplexing and organization of the sample sequences in FASTQ files were performed with bcl2fastq2 software. Quality control and trimming procedures were conducted with FastQC/0.11.8 (12) and TrimGalore/0.4.4 with a stringency of 5, and the results were summarized with MultiQC/1.8 (13). The rRNA content in the samples was assessed with SortMeRNA/4.3.6 (14).

To reduce file size and enable sharing in an online repository, the FASTQ files were depleted of human reads using a two-step method to detect human reads with the Kraken2/2.1.1 (15) and Bowtie2/2.4.4 (16). Reads that were classified as *Homo sapiens* by Kraken2 using a premade Refseq index from the Standard collection (dated 12 September 2022) were removed using seqtk/version1.2 in the first step of depletion. In the second depletion, reads aligning to the human genome assembly GRCh37 (hg19) with Bowtie2/2.4.4 were removed with Samtools/1.15.1 (16). The human-depleted reads were taxonomically assigned again with Kraken2/2.1.1 and the Standard database (dated 12 September 2022), comprising archaea, bacteria, viral, plasmid, and human sequences. The relative abundance of bacterial species at the species level in the samples was estimated with the Bracken tool in the Kraken2 package.

## Creation of the *H. influenzae* core and pan genomes

Core and pan genomes of the 15 *H. influenzae* reference genomes in the KEGG GENOME database (17) (Table S1) were created with Roary/15.36.42 after annotation with Prokka/1.11 (18, 19). The core genome was defined as genes present in all 15 reference strains, and the pan genome was considered the full list of genes present in one or more of these strains. Subsequent use of the phrases "pan genome" and "core genome" refers to these two specific sets of genes. Genes that were assigned gene names in the form of "group_??" or gene names that could not be recognized by the PANTHER database were then renamed by translating them into an amino acid sequence with EMBOSS and aligning them to protein sequences from the *H. influenza* strains in UniProt (20) (2021-10-13) using BLAST, version 2.7.1. We filtered matches with e-value <0.0001 and identity% <95%. Genes in the core genome that were still unidentified were manually identified by performing a BLAST search on the nucleotide sequence and adding (primarily) gene names or (secondarily) loci from *H. influenzae* Rd KW20 when available. If the gene could not be identified in the Rd KW20 genome, *H. influenzae* PittII loci were used.

## Pseudo-alignment of *H. influenzae* to the core and pan genomes

To measure transcript reads from the RNA-seq analysis that could be assigned to *H. influenzae,* reads were pseudo-aligned with Kallisto/0.46.2 using the established core and pan genomes (21). The counts of the aligned reads for each sample were then used in the downstream analysis of differential gene expression.

## Analysis of differential gene expression

The DESeq2/1.32.0 package (22, 23) in R/4.2.0 (24) was used to analyze differential gene expression and perform the statistical analysis. Kallisto output data were transformed to gene counts with tximport/1.20.0. Genes with fewer than 10 counts in total (across all samples) were filtered. Genes with zero read counts in a sample were treated as zeros for this sample. $P$ values and adjusted $P$ values were calculated by Wald test and adjusted by Benjamini-Hochberg method to assess the significance of differences in mean expression between *in vivo* (clinical) and *in vitro* (lab-cultured) samples. Differentially expressed genes (DEGs) were defined as genes expressed with a significant (adjusted $P$ value ≤ 0.05) log2 fold-change (logFC) of ≥1 in bacteria in the clinical *in vivo* samples compared

to the bacteria cultured *in vitro*. To reduce dimensionality and visualize the data, principal component analysis (PCA) was performed after variance-stabilizing transformation. Clustered heatmaps were generated by Pearson correlation and by Ward's method on variance-stabilized counts. Plots were created in R using the DESeq2, pheatmap, EnhancedVolcano and ggplot2 packages.

## Gene ontology enrichment analysis

Gene ontology (GO) classification of all genes present in the core genome was performed using the PANTHER classification system, version 16 (25). Analysis of statistical overrepresentation and enrichment was done using tools provided in the topGO package (26). When calculating statistical fold-enrichment of GO terms, the DEGs were divided into upregulated and downregulated genes, comparing the respective groups with the rest of the core genome. The *weight01* algorithm and Fisher statistics were used for this analysis in topGO, and a *P* value of ≤0.05 was chosen as the cutoff for significance. The *weight01* algorithm is a mixture of the *elim* and *weight* algorithms, which take GO hierarchies into account when calculating *P* values, leading to a more conservative result compared with the classical algorithm (27). Since this method accounts for GO topology, the tests are not independent, rendering corrective measures for multiple testing inapplicable.

## RESULTS

### Study participants and *in vivo* and *in vitro* samples

Out of the 102 participants recruited in the study (*N* = 57 and *N* = 45 from the emergency department and ICU, respectively), lower respiratory tract samples from 18 participants contained culturable *H. influenzae* (Fig. 1). The clinical and demographic data of the study participants are summarized in Table 1; the details of each study participant are listed in Table S2. RNA was extracted from all 18 samples, and samples from which RNA of acceptable quality could be extracted and libraries could be generated were sequenced at between 215 and 509 M reads per sample (mean 341.5 M reads). The resulting reads were taxonomically classified, and samples with less than 1% *H. influenzae* reads or that contained other species in the *Haemophilus* genus were excluded. Eight samples were thus selected for downstream bioinformatic analysis. On average, these samples contained 46.5% (SD 22.9) bacterial reads and 9.2% (SD 6.3) reads assigned to *H. influenzae* in the taxonomic analysis. Details of the RNA sequencing are available in Table S3, and the taxonomic classification of the samples is available in Supplementary file 3.

In addition to respiratory tract samples for RNA analysis, the clinical microbiology laboratory provided us with the corresponding cultured *H. influenzae* isolates from six participants (Fig. 1). Two separate colonies of *H. influenzae* were randomly selected from each cultured sample and one reference strain that was included for comparison (NTHi 3655). These were cultured to the exponential phase in rich broth media, followed by RNA extraction and sequencing with 15.7–22.0 M reads (average 18.5 M reads). Depletion of rRNA in the samples was evaluated in all sequence files and was found to be more efficient in the human metagenomic samples than in the samples with only bacterial RNA. The mean rRNA content in the *in vivo* samples was 27.1% (SD 11.8) compared with 83.4% (SD 13.9) in the *in vitro* samples. Genomic DNA from the cultured bacterial strains was also sequenced, and MLSTs and serotypes were determined *in silico* from the assembled genomes (Table 2).

In total, nine study subjects were included in the study, together yielding eight *in vivo* samples and six *in vitro* isolates that were used in downstream bioinformatic analyses.

### Core and pan genomes

To create a reference against which the RNA-sequencing reads could be counted, a pan genome was created from the 15 *H. influenzae* genome assemblies in the KEGG database. These include genomes from some of the most extensively studied reference

**TABLE 1** Demographic data of included study subjects[a]

| Characteristic | Study population ($N = 9$) |
|---|---|
| Mean age (SD), year | 67.9 (12.3) |
| Sex, M | 5 (56) |
| Previous medical conditions and comorbidities | |
| *COPD/Asthma/Emphysema* | 6 (67) |
| *Diabetes mellitus* | 2 (22) |
| *Immunodeficiency* | 3 (33) |
| *Smoking history* | 6 (67) |
| Antibiotic treatment last 5 days | 3 (33) |
| Clinical data at enrollment | |
| *Purulent sputum* | 6 (67) |
| *Temp > 38°C or <35°C last 24 h* | 5 (56) |
| *New pulmonary infiltrate* | 7 (78) |
| *Mean pneumonia-symptom duration (SD), days* | 3.2 (3.5) |
| *Median qSOFA total points* (range) | 1 (0–3) |
| *Mean plasma-CRP (SD), mg/L* | 211 (123.9) |
| *Mean white blood cell count (SD), $10^9$/L* | 13.9 (3.6) |
| Admission type | |
| *Not admitted* | 1 (11) |
| *Ward* | 5 (56) |
| *ICU* | 3 (33) |
| ICU outcomes | |
| *Mean FiO2 (SD), %* | 53 (21) |
| *Mean length of ventilator treatment (SD), days* | 1.3 (0.6) |
| 30-day mortality | 2 (22) |

[a]Demographics and clinical data of the study population. Values indicate $n$ (%) unless otherwise specified. Comorbidities, previous and ongoing medications, vital sign assessment, and laboratory values were retrieved from the hospital records in the emergency departments or intensive care units (ICU), and qSOFA scores were calculated from these values. No study subject suffered from cystic fibrosis, bronchiectasis, pulmonary fibrosis, pulmonary cancer, or other tumor disease. Smoking history includes both current ($N = 3$) and previous ($N = 3$) smoking habits; all patients who were admitted to the ICU were current smokers. COPD, chronic obstructive pulmonary disease; qSOFA, quick sepsis-related organ failure assessment.

*H. influenzae* strains. In total, this pan genome consisted of 4,163 genes of which a large proportion were present in only one of the included reference genomes (1,942 genes, 46.6%). From the pan genome, a core genome was extracted based on the 1,072 genes (25.8%) that were present in all included genomes. For each additional strain that was included in the creation of the pan genome, it continued to expand, while the core genome only reduced moderately. This indicates that the genes of the created core genome were well conserved within this bacterial species and provide a robust reference for the gene expression analysis (Fig. S1A-C).

## Gene expression counts

The *H. influenzae* reads from the RNA-sequencing analysis of all respiratory tract *in vivo* samples and laboratory-cultured *in vitro* samples were counted after pseudo-alignment to the created core and pan genomes. On average, 639,067 reads (range 76,357–1,527,132) from the *in vivo* samples pseudo-aligned to the core genome, versus 924,569 (range 126,020–2,289,609) pseudo-aligning to the pan genome. The corresponding average counts for the *in vitro* samples were 829,354 (range 237,372–2,994,321) to the core genome and 1,097,743 (range 316,822–3,763,386) to the pan genome. A total of 1,067 of the core genes and 3,200 of the pan genes were found to be expressed in at least one of the samples included in the present study. Furthermore, 768 of the genes in the core genome (72.0%) were expressed in all samples, while 204 (19.1%) had zero read counts in one sample and 95 (8.9%) had zero read counts in two or more samples. Full read count tables are available in Supplementary file 4.

**TABLE 2**  Overview of samples and bacterial isolates[a]

| Study subject | Clinical sample, *in vivo* condition | Cultured *in vitro* samples | MLST | Clonal complex | Serotype | PubMLST ID |
|---|---|---|---|---|---|---|
| LUIN_26 | *LUIN_26* | *HI_LUIN_26_1* and *HI_LUIN_26_2* | 155 | ST-155 | NTHi | 24543 |
| LUIN_28 | NA | *HI_LUIN_28_1* and *HI_LUIN_28_2* | 266 | ST-266 | NTHi | 24541 |
| LUIN_29 | *LUIN_29* | NA | NA | NA | NA | NA |
| LUIN_31 | *LUIN_31* | NA | NA | NA | NA | NA |
| LUIN_33 | *LUIN_33* | *HI_LUIN_33_1* and *HI_LUIN_33_2* | 103 | ST-11 | NTHi | 24542 |
| MAAK_03 | *MAAK_03* | *HI_MAAK_03_1* and *HI_MAAK_03_2* | 155 | ST-155 | NTHi | 24538 |
| MAIV_24 | *MAIV_24* | *HI_MAIV_24_1* and *HI_MAIV_24_2* | 12 | ST-12 | NTHi | 24539 |
| MAIV_32 | *MAIV_32* | *HI_MAIV_32_1* and *HI_MAIV_32_2* | 2677 | ST-1529 | NTHi | 24540 |
| MAIV_34 | *MAIV_34* | NA | NA | NA | NA | NA |

[a]Overview per study participant of the clinical respiratory tract samples used for *in vivo*-condition analyses and cultured bacterial isolates used for *in vitro*-condition analyses. Further clinical details about each study participant is available in Table S2. Genomic DNA from all cultured isolates were sequenced and assembled *de novo* and assigned to the multi-locus sequencing type (MLST) scheme, clonal complex, and capsule type at pubmlst.org. In addition to the listed samples, the reference strain NTHi 3655 was included in all *in vitro*-condition analyses. Not all samples were available for analysis under all conditions.

## Unsupervised clustering and data visualization

For dimensionality reduction and visualization of the RNA-sequencing data, PCA was performed, including all core and pan genome genes (Fig. 2A and B). In the core genome analysis, the *in vitro* cultured bacterial isolates clustered tightly, regardless of genetic background. The *in vivo* gene expression profile showed higher transcriptional diversity and did not cluster with the corresponding bacterial isolate cultured *in vitro*. In the pan genome PCA, the *in vitro* gene expression profile was less homogeneous, and isolates formed a loose cluster.

The same pattern of clustering by growth condition was seen in a heatmap displaying the logFC of the most variable core genome genes, where *in vivo*-sampled isolates showed more heterogeneity with higher contrast compared with the more homogeneous appearance of lab-grown samples (Fig. 2C). In the pan genome heatmap (Fig. 2D), this pattern was less clear, and one *in vivo* sample (LUIN_33) instead clustered with its lab-cultured counterpart, which is most likely a result of the expression of a gene set not present in all clinical strains in the analysis.

## Differential expression analysis

To identify individual genes that were differentially expressed between *H. influenzae* cells growing in the lung and cells in exponential growth phase in nutrient-rich media, differential gene expression analysis was performed, including subject as a covariate to account for the heterogeneity among bacterial genomes. In the 328 genes of the core genome that were defined as DEGs in this process (30.7%), 130 were upregulated (39.6%) and 198 (60.4%) were downregulated *in vivo*. In total, 488 core genome genes were upregulated *in vivo* (45.7%), while 579 were downregulated (54.3%). Of the 558 identified DEGs in the pan genome (17.4%), 269 and 289 were upregulated and downregulated, respectively (48.2 vs 51.8%). In total, 1,747 genes of the entire pan genome were upregulated (54.6%); 1,453 (45.4%) were downregulated. The logFCs and *P* values of the core genes are visualized in a volcano plot in Fig. 3, and the corresponding volcano plot for the pan genome analysis is available in Fig. S2 (full lists of logFCs/*P* values are presented in Supplementary files 5 to 7).

In general, all analyses of the pan genome displayed high heterogeneity, probably due to the fact that almost half of the genes in the pan genome were present in only

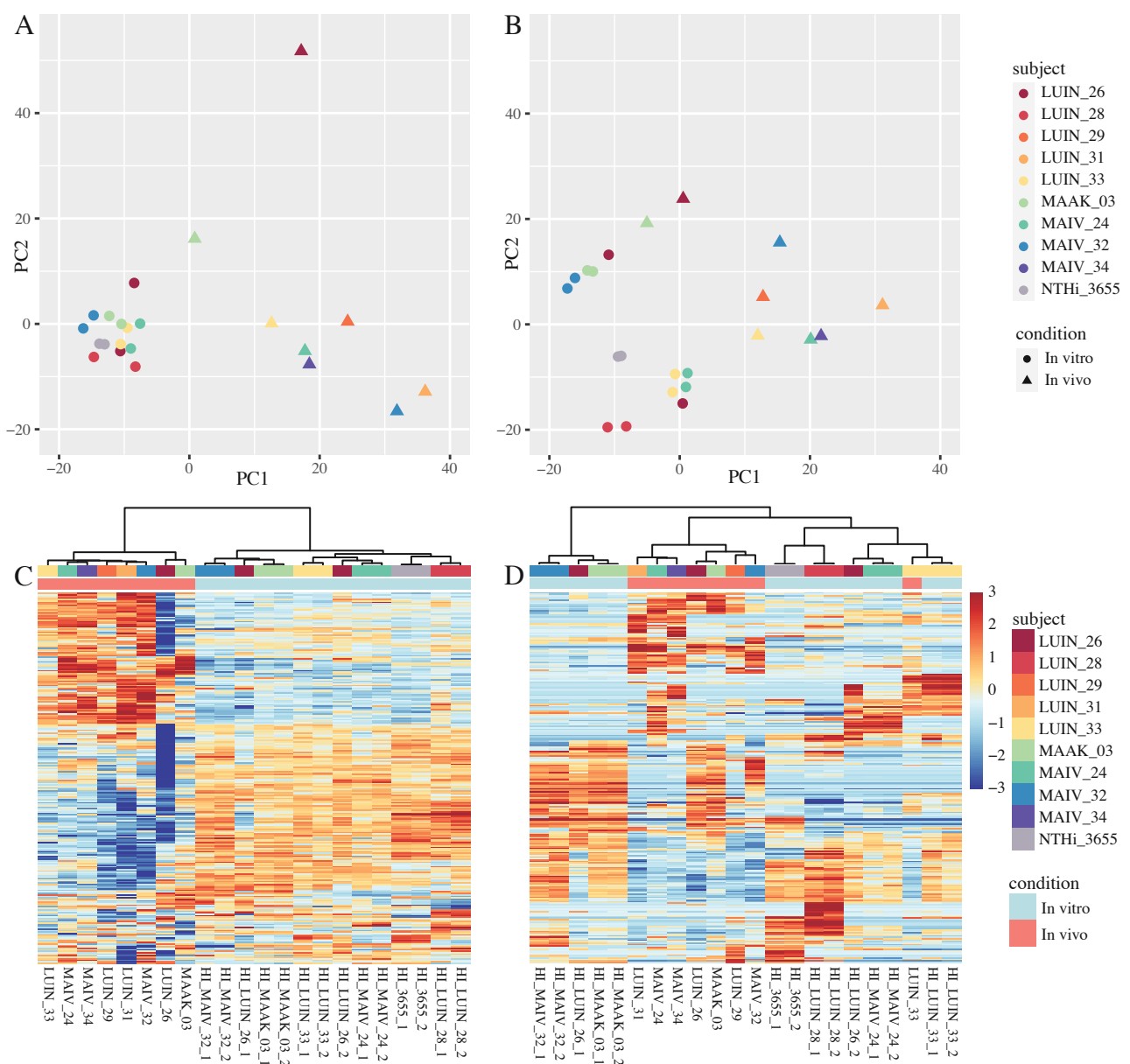

**FIG 2** PCA plot of expression of all core (A) and pan genes (B) and heatmaps displaying the normalized counts of the top 250 most variable genes included in the core (C) and pan genomes (D). In the PCA plot, *in vitro* isolates are represented by dots, and *in vivo* isolates are indicated by triangles; each color represents one study subject. In the core genome analysis (A), *in vitro* isolates cluster tightly together, while *in vivo* samples show more heterogeneity in their transcriptional profiles (PC1: 32% explained, PC2: 20% explained). In the pan genome analysis (B), this pattern is less obvious but still apparent, although *in vitro* isolates seem to cluster into two groups rather than one (PC1: 22% explained, PC2: 19% explained). In the heatmap of the top variable core genome genes (C), using unsupervised clustering, the *in vitro* and the *in vivo* isolates cluster with their respective culture condition rather than with the corresponding isolate. Genes of *in vivo* isolates show more diversity with higher contrast in this analysis, while the transcriptome of *in vitro* isolates is more homogeneous in appearance. The legend scale indicates the log-transformed normalized counts, cropped at −3 and +3. In the top 250 variable pan genome genes (D), clustering is more affected by the genetic background of the isolates. Sample names starting with "HI_" are *in vitro*-cultured in duplicates (suffix "_1" and "_2").

one strain. Since the analysis of genes that were present in only one or a small number of strains was unlikely to yield robust results and since the main objective of this study was

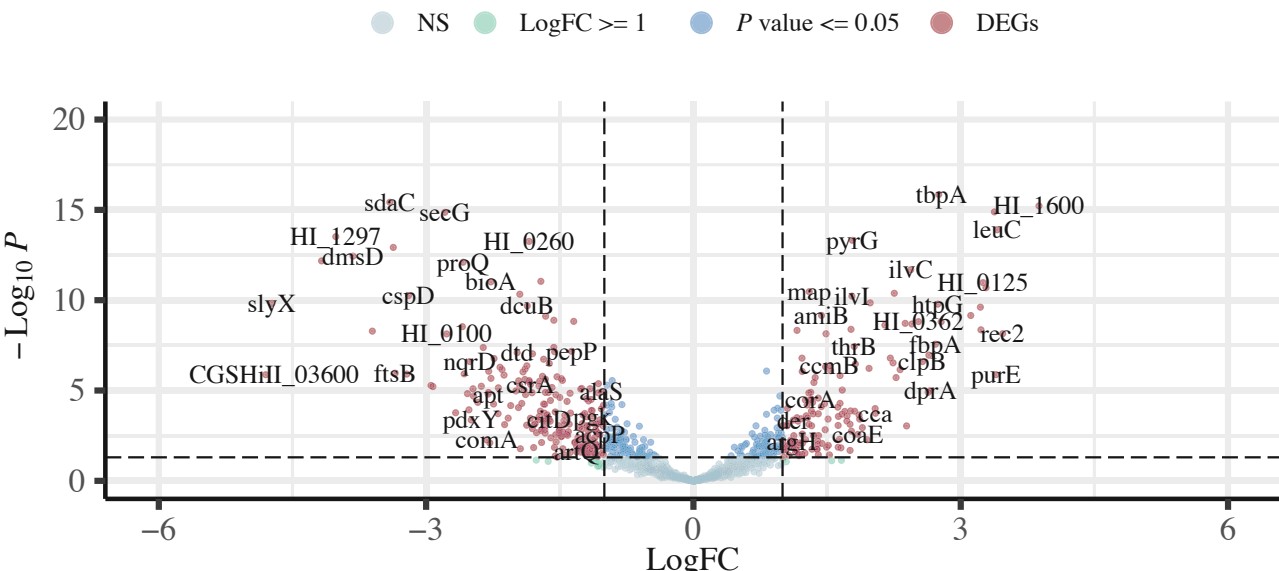

**FIG 3** Volcano plot displaying the dispersion of core genes separated by logFC (x-axis) and *P* value (y-axis, log$_{10}$-transformed and inverted adjusted *P* values). Selected gene names and loci of the most differentially expressed genes are shown in the graph.

to examine the universal biological properties of *H. influenzae*, subsequent analyses were focused on the core genome.

Despite large variations in host and clinical parameters, many genes exhibited large differences in expression level between groups but stable within-group expression, as visualized in a heatmap and a plot of the expression of the 30 core genes showing the highest absolute logFC changes between the two studied conditions (Fig. 4 and 5). The corresponding heatmap for the pan genome is shown in Fig. S3.

Many of the core DEGs with the highest relative expression *in vivo* were involved in similar pathways or related by analogous predicted functions. For instance, the gene products of the neighboring genes *HI_1453* and *HI_1454* were predicted functional partners in the STRING database with *msrAB*, a gene involved in the response to oxidative stress (28). Similarly related with each other were the genes *purE*, *purH,* and *purK* (inosine monophosphate/IMP *de novo* biosynthesis), the genes *rec2* and *dprA* (probably involved in transformation), the genes *tbpA* and *fbpA* (import of transferrin and iron, respectively) and the genes *HI_1599* and *HI_1600* (both with unknown function) (20, 25, 28).

## Enrichment analysis of gene ontology terms related to biological processes

Gene ontology annotations of the genes in the core genome were retrieved from the PANTHER database and the topGO package in R. Of the 1,067 core genes that were expressed in one or more of the samples in the current study, 1,061 (99.4%) could be assigned a GO term. "Biological process" was determined to be the most relevant classification to explore further and use as a comparison between DEGs and the rest of the core genome. A total of 896 (84.4%) annotated genes in the core genome could be assigned a biological process and were included in this analysis. The assigned biological processes of all genes in the analysis are found in Supplementary file 8.

Significantly enriched GO terms, compared with the rest of the core genome, were determined separately for the downregulated and upregulated DEGs; 11 were found among the downregulated DEGs, versus 13 in the upregulated DEGs. All significantly enriched biological process terms are presented in Tables 3 and 4. Lists of the 50 most

## Expression of top 30 DEGs

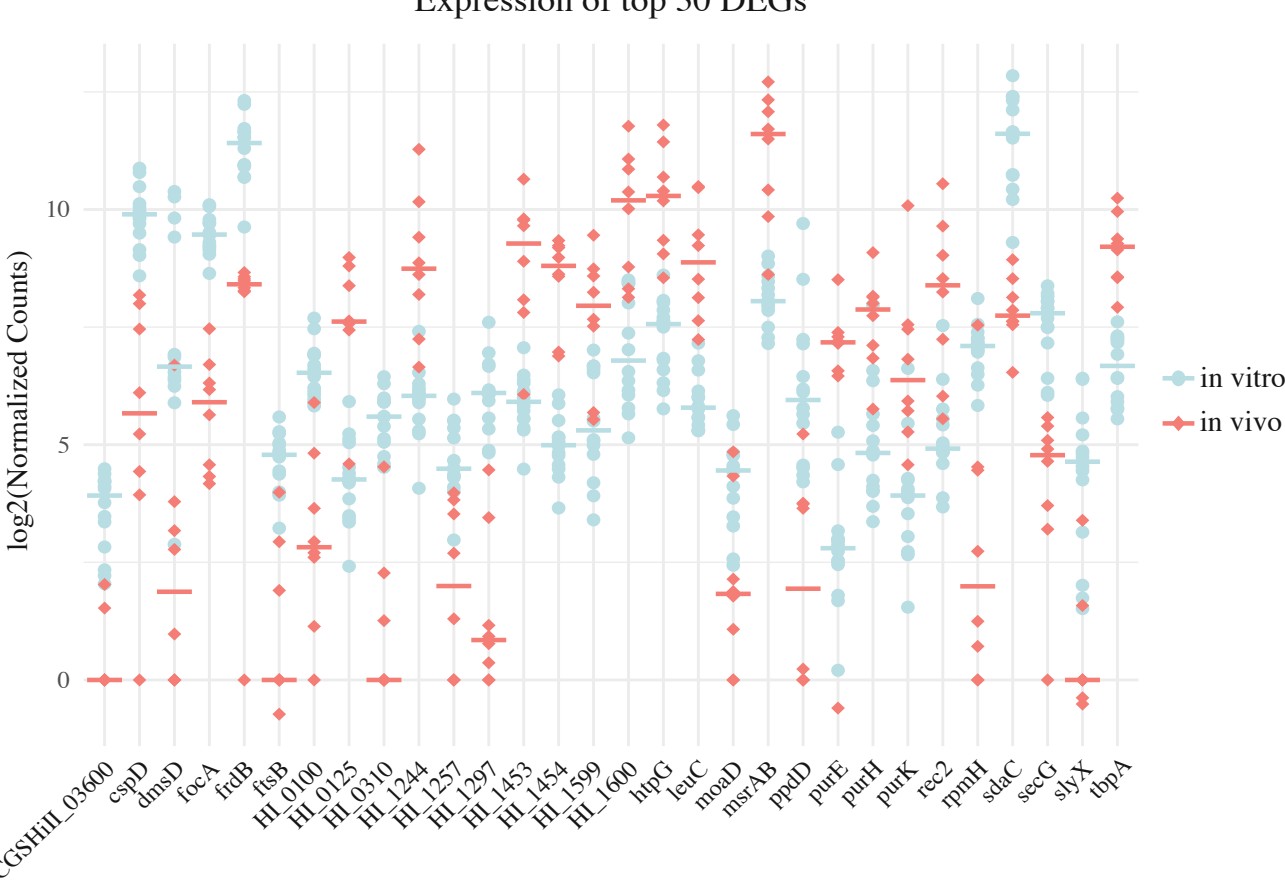

**FIG 4** Dot plot showing log2-transformed normalized read counts of *in vivo* and *in vitro* isolates featuring the expression of the top 30 differentially expressed genes (i.e., showing the highest absolute logFC). Lines indicate the median count value for each group. For visualization purposes, +1 was added to count values of 0 before log transformation.

highly enriched biological processes for the downregulated and upregulated DEGs are available in Tables S4 and S5.

The enriched GO terms for biological processes in the downregulated DEGs were related mainly to *H. influenzae* energy production and metabolic pathways. The processes of glycolysis, gluconeogenesis, the tricarboxylic acid (TCA) cycle, and the metabolism of some of its TCA intermediates, were all enriched in these downregulated DEGs. We also examined the change in expression of the 23 genes in our core genome that were directly involved in the respiratory chain. Twenty of them had lower relative expression *in vivo,* and 16 were downregulated DEGs, indicating lower metabolic activity for *H. influenzae* cells in the human lung compared with cells in rich media. The genes of the TCA cycle, glycolytic process, and pentose-phosphate shunt of *H. influenzae* are shown in schematics in Fig. 6, and Fig. 7A displays the normalized expression of all genes assigned to the TCA cycle. The GO terms "iron transport" and "intracellular sequestering" were also found to be enriched in the downregulated DEGs (Fig. 7B). We found no other iron-related process that differed significantly between DEGs and the rest of the core genome, including iron homeostasis (GO:0055072 or GO:0006879).

The GO terms for biological processes that were enriched in the upregulated DEGs were related to the biosynthesis of purines/pyrimidines and other components of DNA and RNA; biosynthesis of the branched-chain amino acids leucine, valine, and isoleucine; DNA unwinding and transcription; response to heat; phosphate transmembrane transport; and transport of molybdate (Mo). The *de novo* biosynthetic process of the

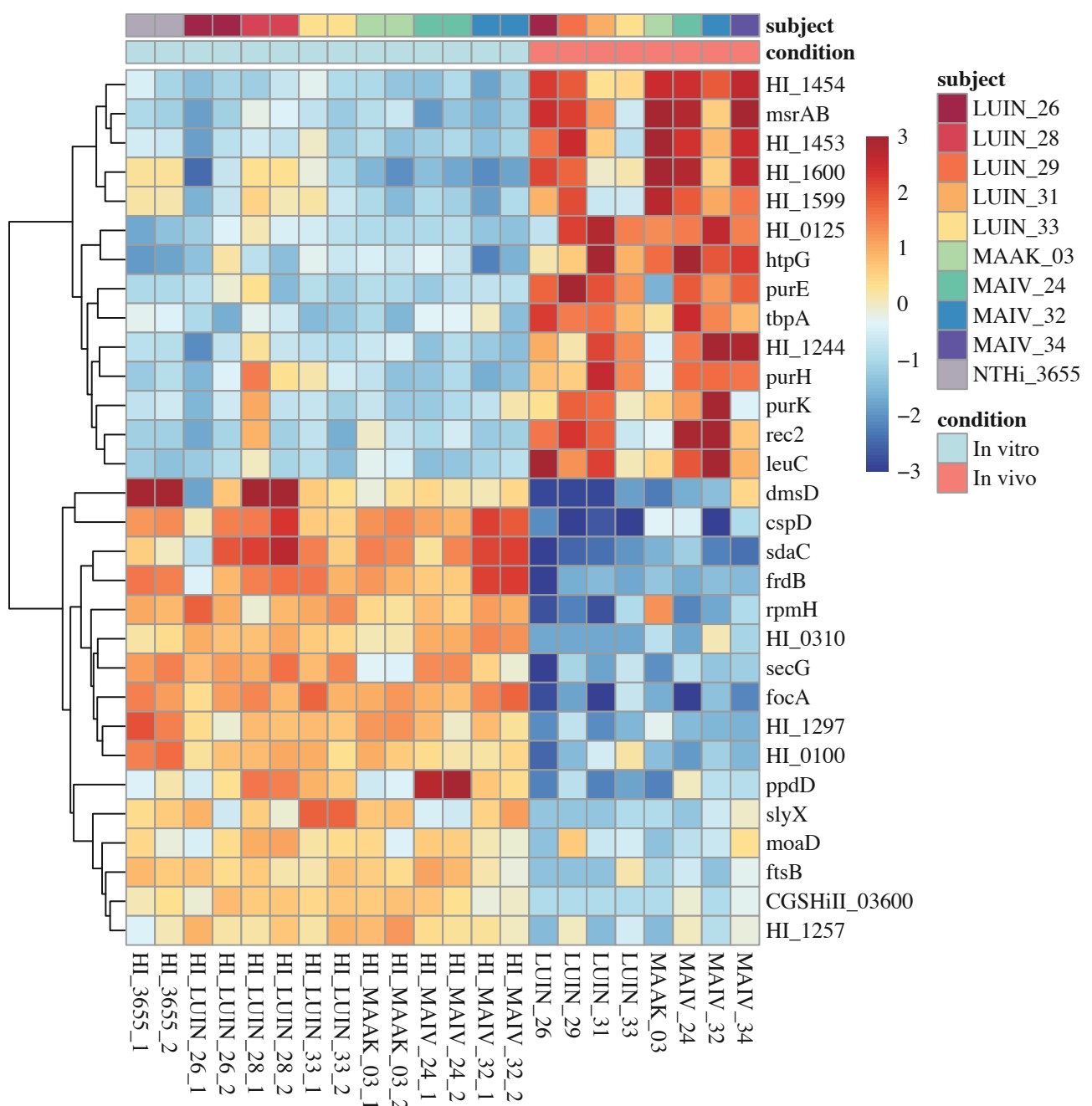

**FIG 5** Heatmap displaying the top 30 differentially expressed genes. Fourteen genes showed higher (red) expression *in vivo* versus 16 that were lower (blue). The legend scale indicates the log-transformed normalized counts, cropped at −3 and +3.

purine precursor inosine monophosphate (IMP) was the most highly enriched GO term overall, with the majority of purine regulon genes (30) being upregulated DEGs (Fig. 7C). Despite *msrAB* being among the most highly upregulated DEGs and being assigned to the GO term "response to oxidative stress" (GO:0006979), no significant difference was seen in the expression of genes involved in stress response pathways, such as oxidative stress, response to different chemicals, and response to DNA damage.

All of the observed changes in the expression of genes that were involved in molybdate transport occurred in *modABC* operon genes, which are responsible for the influx of Mo into the bacterial cell (31, 32). These were significantly upregulated *in vivo*

**TABLE 3** Significantly enriched GO terms within "biological process" for the downregulated DEGs[a]

| Gene ontology ID | Term | Annotated (n) | Significant (n) | Expected (n) | Fold-enrichment | Fisher's exact test |
|---|---|---|---|---|---|---|
| GO:0006108 | Malate metabolic process | 3 | 3 | 0.5 | 5.66 | 0.005 |
| GO:0006094 | Gluconeogenesis | 7 | 4 | 1.2 | 3.25 | 0.020 |
| GO:0006826 | Iron ion transport | 8 | 4 | 1.4 | 2.86 | 0.030 |
| GO:0046854 | Phosphatidylinositol phosphate biosynthetic process | 2 | 2 | 0.3 | 5.71 | 0.030 |
| GO:0046855 | Inositol phosphate dephosphorylation | 2 | 2 | 0.3 | 5.71 | 0.030 |
| GO:0019646 | Aerobic electron transport chain | 2 | 2 | 0.3 | 5.71 | 0.030 |
| GO:0006880 | Intracellular sequestering of iron ion | 2 | 2 | 0.3 | 5.71 | 0.030 |
| GO:0006106 | Fumarate metabolic process | 2 | 2 | 0.3 | 5.71 | 0.030 |
| GO:0006654 | Phosphatidic acid biosynthetic process | 2 | 2 | 0.3 | 5.71 | 0.030 |
| GO:0006099 | Tricarboxylic acid cycle | 8 | 4 | 1.4 | 2.86 | 0.035 |
| GO:0006096 | Glycolytic process | 13 | 6 | 2.3 | 2.63 | 0.041 |

[a]All significantly enriched GO terms within "biological process" for the downregulated DEGs are listed. "Annotated" denotes the total number (n) of genes present in the core genome annotated to each term. "Significant" denotes the number of these genes that are DEGs, while "expected" denotes the expected number of annotated genes to be present in DEGs. Fold-enrichment was calculated as "significant" divided by "expected."

(logFC 2.32, 1.67, and 1.98, respectively). *modE* was the only gene in this operon that was not an upregulated DEG, consistent with this protein functioning as a transcriptional regulator, acting as a Mo-dependent repressor of this pathway (32–34). The Mo-molyb-dopterin cofactor biosynthetic process was not included among significant GO terms, but the gene expression of many enzymes that participate in this pathway was lower *in vivo* (seven out of nine genes having a negative logFC, ranging from −2.95 to −0.47), one of which was among the 16 genes showing the highest decrease in expression *in vivo* (*moaD*; Fig. 4 and 5).

## DISCUSSION

The primary aim of the present study was to describe the *H. influenzae* transcriptome *in vivo* in the lung of patients with pneumonia compared with bacterial cells cultured under typical laboratory conditions *in vitro*—i.e., shaking cultures in rich media, stopped at the exponential phase. In this first clinical *in vivo* transcriptome sequencing study of *H. influenzae* gene expression, we found several interesting themes worth exploring further.

**TABLE 4** Significantly enriched GO terms within "biological process" for the upregulated DEGs[a]

| Gene ontology ID | Term | Annotated (n) | Significant (n) | Expected (n) | Fold-enrichment | Fisher's exact test |
|---|---|---|---|---|---|---|
| GO:0006189 | "*de novo*" IMP biosynthetic process | 8 | 7 | 1.0 | 7.22 | 0.000 |
| GO:0009098 | Leucine biosynthetic process | 5 | 4 | 0.6 | 6.56 | 0.001 |
| GO:0032784 | Regulation of DNA-templated transcription, elongation | 3 | 3 | 0.4 | 8.33 | 0.002 |
| GO:0015689 | Molybdate ion transport | 4 | 3 | 0.5 | 6.12 | 0.006 |
| GO:0009113 | Purine nucleobase biosynthetic process | 4 | 3 | 0.5 | 6.12 | 0.014 |
| GO:0009099 | Valine biosynthetic process | 5 | 3 | 0.6 | 4.92 | 0.014 |
| GO:0044210 | "*de novo*" CTP biosynthetic process | 2 | 2 | 0.2 | 8.33 | 0.014 |
| GO:0032508 | DNA duplex unwinding | 9 | 4 | 1.1 | 3.67 | 0.026 |
| GO:0006164 | Purine nucleotide biosynthetic process | 36 | 11 | 4.4 | 2.51 | 0.036 |
| GO:0009116 | Nucleoside metabolic process | 15 | 5 | 1.8 | 2.75 | 0.039 |
| GO:0009408 | Response to heat | 5 | 3 | 0.6 | 4.92 | 0.039 |
| GO:0035435 | Phosphate ion transmembrane transport | 3 | 2 | 0.4 | 5.56 | 0.040 |
| GO:0009097 | Isoleucine biosynthetic process | 7 | 3 | 0.8 | 3.53 | 0.041 |

[a]All significantly enriched GO terms within "biological process" for the upregulated DEGs are listed. "Annotated" denotes the total number (n) of genes present in the core genome annotated to each term. "Significant" denotes the number of these genes that are DEGs, while "expected" denotes the expected number of annotated genes to be present in DEGs. Fold-enrichment was calculated as "significant" divided by "expected."

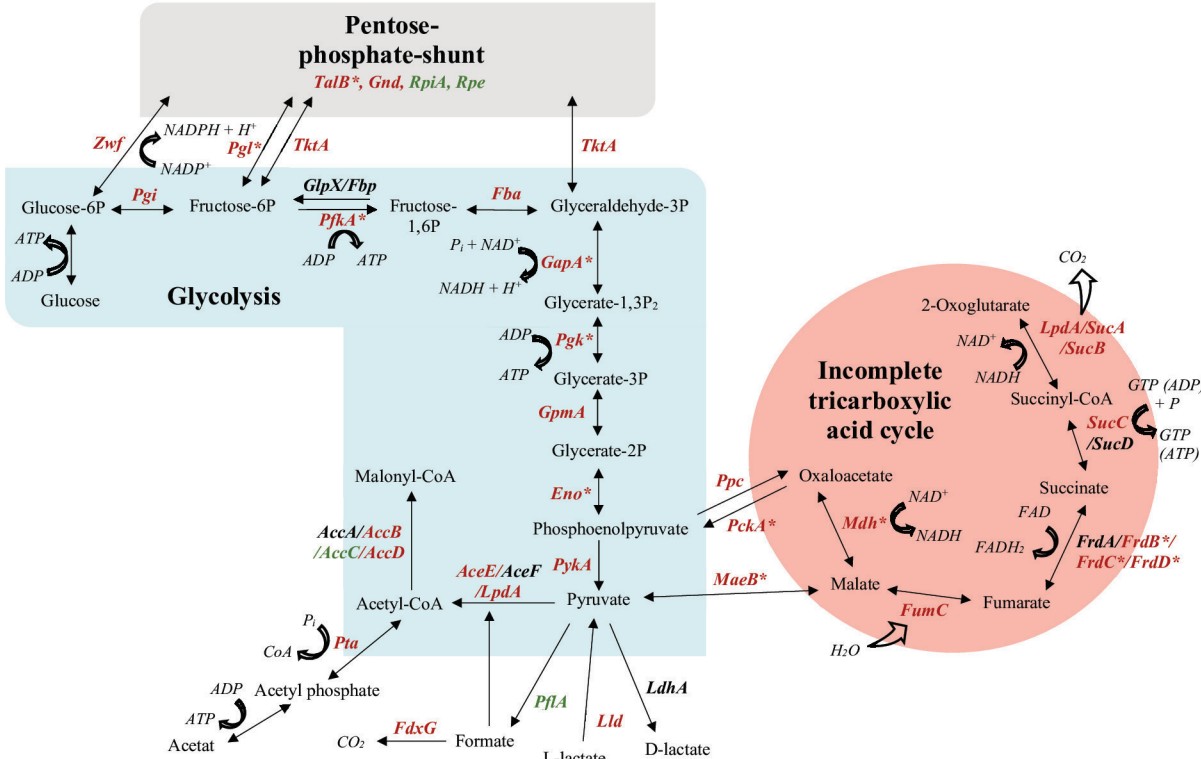

**FIG 6** Schematics of genes and reactions involved in the tricarboxylic acid cycle, glycolysis, and the pentose phosphate pathway of *H. influenzae*. Color of gene names indicates the direction of the change in gene expression in our material (green: upregulated *in vivo*, red: downregulated *in vivo*, black: not present in core genome). *DEGs: significant ($P \le 0.05$) logFC > 1. Based on figure from López-López et al. 2020 (29).

To enable mapping and counting of the RNA sequencing reads, pan and core *H. influenzae* genomes were created: 1,072 genes were present in all 15 reference strains and formed the core genome, a slightly lower number than in previously reported *H. influenzae* core genomes (4). We included both encapsulated and NTHi reference strains of different sequence types to ensure that we analyzed conserved biological functions of *H. influenzae*, regardless of the presence of capsular or genetic variation. The difference between our core genome and previous works is likely to be explained by variations in methods and the choice of strains that were analyzed. By adding more strains to the analysis, we could have limited the core genome further, although we found that each additional strain had a limited effect on the size of the core genome. However, the pan genome continued to expand for each additional strain, as expected for a bacterial species with an open genome. Hence, the level of complexity in terms of annotation that was created by adding more strains increases with each additional strain.

For the current study, by limiting the core and pan genomes to the 15 references in the KEGG database, we were able to create a representative core genome to perform a meaningful analysis of conserved genes required for basic cellular functions while limiting the size of the pan genome. Choosing more similar or matching *H. influenzae* strains to those included in the study or limiting them to specific types might have resulted in core and pan genomes of a different size. As expected, many genes that were related to antibiotic resistance and virulence were not included in the core genome, as they were not shared between all strains that were used for the generation of this gene set. Similarly, only two clinical isolates (LUIN_31 and LUIN_33) carried and expressed the *bla* gene, which encodes the beta-lactamase TEM-1.

Furthermore, interesting discoveries could have been made through analysis of the pan rather than core genome or by strict pairwise comparison between clinical strains and their lab-grown counterparts, which include the entire chromosome and plasmids.

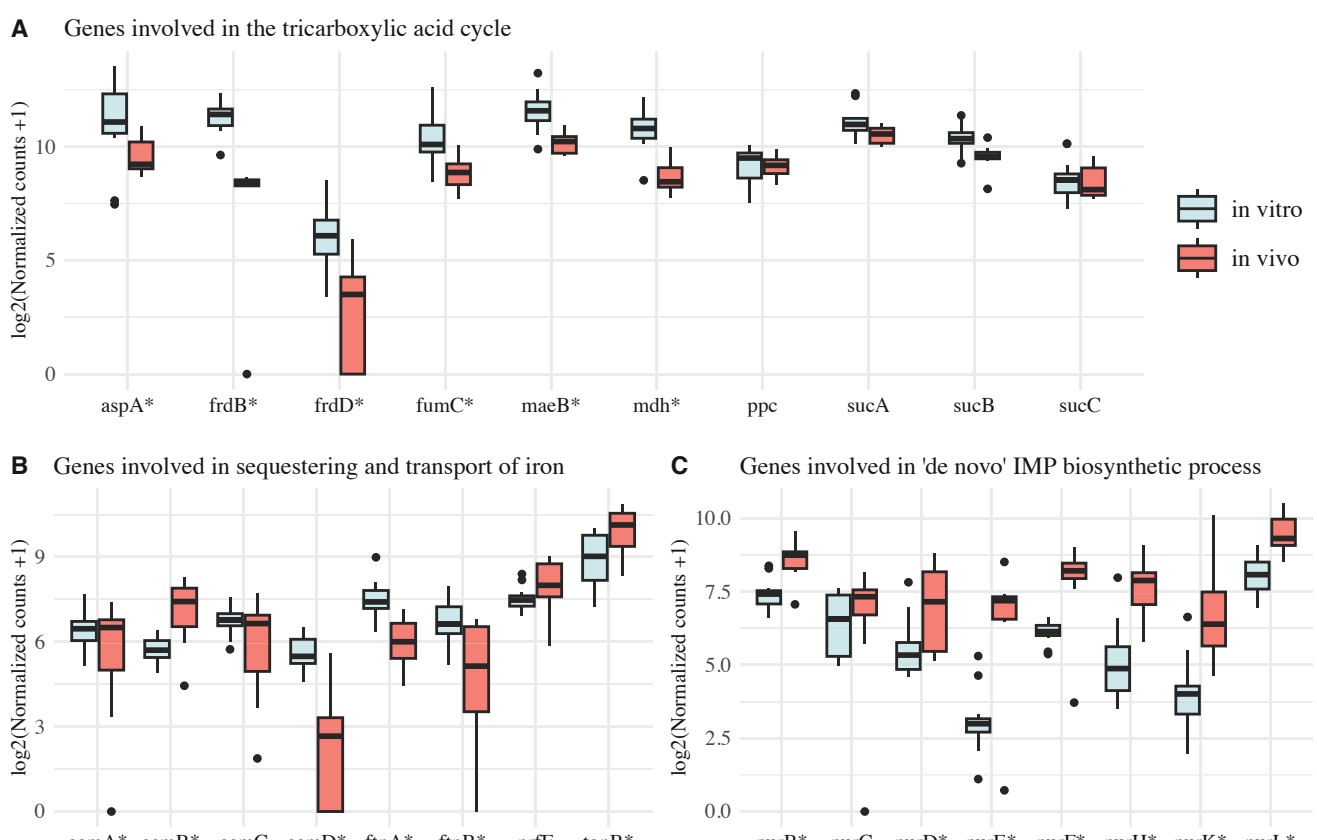

**FIG 7** The expression of all core genes (log2-normalized counts + 1) belonging to the following biological processes: significant processes relevant for the TCA cycle (A: enriched in the downregulated DEGs), iron sequestering and transport (B: enriched in the downregulated DEGs), and "*de novo*" IMP biosynthetic process (C: enriched in the upregulated DEGs).

However, the lack of replicates would complicate the statistical analysis in a paired-sample approach, and the large genetic variation complicates the use of the pan genome as a reference against which the reads are counted. In our material, 30.7% of core genome genes were significantly differentially expressed. This was lower than in a previous report comparing the effect of *in vitro* host-bacterial cell interactions on the gene expression profile of a single strain of NTHi, in which 1,068 of the 1,877 *H. influenzae* genes (56.9%) were differentially expressed (6).

Our most notable observation was that bacteria in the human lung have a transcriptomic signature that is distinct from bacteria grown *in vitro* under controlled laboratory conditions. The isolates that were cultured in a controlled environment clustered tightly, regardless of genetic background. The greater importance of the growth environment to a transcriptomic signature versus the genetic background was previously reported by Aziz et al. (35). These authors reported modest transcriptomic differences for NTHi isolates that were adapted either to the nasopharynx or to the lower respiratory airways and then cultured under the same conditions. Similarly, genetically diverse *Pseudomonas aeruginosa* isolates collected from human lung exhibited similar gene expression profiles when cultured *in vitro* (36).

Numerous factors might have influenced the more heterogeneous gene expression pattern of the *in vivo*-growing *H. influenzae* in the present study, such as the large variations in host factors (e.g., temperature, ongoing treatment, severity of disease, immunologic status, presence of other bacterial species), compared with the homogeneous culture procedures *in vitro*. *In vitro* cultures from sample LUIN_26 indicate the

presence of possible polyclonality, which also may have affected the *in vivo* transcriptomes (however this was not confirmed by DNA sequencing where the isolates had the same MLST). Other than this exception, the *in vivo* samples cluster more closely, and we conclude that bacterial core genome expression is more dependent on environmental conditions than on genetic background.

The gene ontology-based analysis of the core genome transcripts revealed vast differences in central metabolism between growth conditions, most notably the tricarboxylic acid cycle. The TCA cycle in many *Haemophilus* spp. is known to be incomplete due to the lack of aconitase, citrate synthase, isocitrate, and succinate dehydrogenase, rendering the organism prone to anaerobic respiration that is highly dependent on malate dehydrogenase (3, 37–39). In our analysis, several critical genes that are involved in the TCA cycle were significantly downregulated in the clinical isolates. A plausible explanation for the general downregulation of the TCA cycle *in vivo* is that the bacterial cells are in a more dormant and less metabolically active state during clinical infection. This may be related to a stationary or biofilm growth mode (40); the lung constituting a more anaerobic environment; or alterations in energy production based on limitations in the supply of proteins, sugars, and fats. Our findings are also consistent with those of Cornforth et al. who reported a general decrease in the activity of the TCA cycle in *P. aeruginosa* during human infection (41). It should be noted that GO-term based biological interpretations are based on publicly available gene ontology annotations, and the quality of the data is dependent on the quality of the annotation. No biological process could be found for in PANTHER for 165 genes in the present study.

Two genes that are important for competence, *dprA* and *rec2* (42), were among the genes that had the highest relative expression *in vivo*. The natural competence of *H. influenzae* is recognized as having nutritional function, since only 10% to 15% of obtained exogenous DNA is used for homologous recombination during transformation (43), while most nucleic acids are probably catabolized and used in *de novo* DNA synthesis (44, 45). High amounts of free DNA can be found in the mucus of both healthy and diseased human respiratory airways and are available for uptake by the bacterial cells (46, 47). The expression of competence-associated genes is influenced by nutritional signals (CRP-cAMP levels, the phosphotransferase-system, and purine depletion) (44, 48–51). DNA uptake is also highly increased when isolates that are in the exponential phase are transferred from nutrient-rich media to starvation media (52). Thus, we would expect that genes that are involved in competence are induced in the human respiratory tract, and not in supplemented BHI broth.

Several processes that primarily concern replication and transcription were enriched in the *in vivo* upregulated DEGs. They constituted 7 of the 13 most highly enriched GO terms, with *de novo* biosynthesis of purine IMP being the most enriched term—a surprising finding, since the isolates that were cultured *in vitro* were analyzed during the exponential growth phase. Bacteria living in a human host would have restricted availability to pre-synthesized nucleotides and nucleobases, necessitating increased formation of fundamental components of nucleic acids *de novo*. Interestingly, genes that are involved in purine formation have been implicated as virulence determinants in a number of pathogenic bacteria, in addition to *H. influenzae*, including many species of which the respiratory system is the primary niche (53–63). Moreover, a study with attenuated mutant strains in a murine model identified several enzymes that are required for the synthesis of IMP as being essential for *H. influenzae* infection of the lung (64).

Iron availability is limited by the human host as a defense against infectious organisms through utilization of iron-sequestering proteins, such as lactoferrin in epithelial cells; regulation of extracellular iron concentration; and other mechanisms of nutritional immunity (65). We found several genes that are involved in sequestering and transporting of iron to be significantly more expressed in lab-grown isolates, most likely because of relative iron repletion *in vitro*. However, two transporters of iron—the transferrin-binding protein TbpA and the ferric-binding protein FbpA—were among the

most highly upregulated DEGs. Many obligate human pathogens, such as *H. influenzae*, *Neisseria meningitidis,* and *Neisseria gonorrhoeae*, possess the ability to scavenge transferrin-bound iron from their host with transferrin-binding proteins analogous to TbpA, as opposed to the usage of siderophores (66–68). Both FbpA, acting as nodal point in iron transport (69), as well as TbpA, have been described to be upregulated in iron-/heme-restricted surroundings and, derived from animal models, *in vivo* compared to *in vitro* (8, 70). The expression of *tbpA* is also known to be hampered by high levels of hemin (71), which would explain the reduced relative expression *in vitro* observed by us.

The transport of molybdate, a fundamental cofactor of several important prokaryotic redox enzymes (72–76), was another pathway that was significantly enriched in the *in vivo* upregulated DEGs. However, several genes partaking in Mo-molybdopterin cofactor biosynthesis were downregulated *in vivo*. Molybdate availability is limited in the lungs (77), making effective Mo uptake essential. Defective molybdoenzymes have been associated with reduced virulence of various pathogenic bacteria, such as *Escherichia coli*, *P. aeruginosa*, and *Mycobacterium tuberculosis*, and for numerous species, non-functional Mo uptake systems have been shown to create phenotypes with reduced fitness or survival (72). Our findings indicate that while the scavenging of molybdate is intensified *in vivo*, the substrate shortage limits metabolic pathways that require this trace element.

To minimize oxidative stress, the reduction of methionine sulfoxide (MetSO), a product of the oxygenation of small molecules, is essential for bacterial pathogens (78). MsrAB, a MetSO reductase, was highly upregulated *in vivo* in our material (logFC 3.38, $P < 0.001$; included in the 30 most differentially expressed genes). *H. influenzae* isolates have been observed to show increased expression of the *msrAB* gene when exposed to reactive oxygen species that are commonly produced by its human host (79). Lack of the gene also diminishes survival in biofilm and lowers its invasive potential *in vitro* (80). Our results indicate that transcription of *msrAB* is important for persistence of *H. influenzae* in the human respiratory system.

Another MetSO-reducing enzyme, DMSO reductase, has been reported in previous studies to be essential for *H. influenzae* virulence, as mutant strains lacking the *dmsA* gene showed reduced fitness compared with the wild type (81). Remarkably, we found the DMSO reductase-involved genes *dmsA* and *dmsD* to be highly downregulated *in vivo*. One probable explanation for this is that DMSO reductase is a molybdopterin-dependent enzyme, and as noted above, access to molybdate would likely be restricted in the lung compared to in rich growth medium.

Research on *in vivo* gene expression is known for its methodological difficulties, primarily since mRNA is fragile and rapidly turned over. We have taken extensive measures to report the gene expression profile of bacteria living in the human lower respiratory tract, collecting samples bedside to preserve the RNA. The lab-grown cells were harvested after a short centrifugation step, but we cannot exclude the possibility that the RNA was slightly affected by this. Earlier reports have proposed differences in *H. influenzae* metabolic processes to depend on the culture media (29, 82, 83), although extensive studies on alterations in bacterial gene expression between laboratory models are lacking (5).

Despite this precaution and the use of a combination of enzymatic and mechanical cell disruption to maximize the yield, both the amount and quality of the extracted RNA from clinical samples were low, causing sequencing artifacts such as duplicates that needed to be filtered bioinformatically. Sputum can be assumed to contain large amounts of RNases, which may explain the low quality of the extracted RNA, and we were not able to extract RNA with acceptable quality for sequencing from several of the 18 samples that were originally reported to contain culturable *H. influenzae*. Another technical problem we encountered was that the depletion of bacterial rRNA was inefficient. Because of this, the number of reads that could be pseudo-aligned to the *H. influenzae* references in the *in vitro* versus the *in vivo* samples was only marginally lower. A larger difference could have introduced a bias in the statistical analysis.

We counted reads by pseudo-alignment to transcript indices, based on the core and pan genomes. This approach is resource-efficient, which is relevant, due to the large sequence output files, but compared with alignment-based methods, a variability is introduced. Since allelic variation is high among *H. influenzae* strains, some reads may have been wrongly pseudo-aligned, introducing noise into the analysis. The majority of sequenced reads in the clinical samples was human in origin, and among the bacterial reads, a large proportion could be taxonomically assigned to bacterial species other than *H. influenzae*. It cannot be excluded that some reads originating from other bacterial species that express similar genes were wrongly pseudo-aligned to *H. influenzae*. To avoid bias, we therefore used a conservative approach by excluding samples that contained other species in the *Haemophilus* genus. We also excluded samples with less than 1% reads assigned to *H. influenzae* in the taxonomic analysis to avoid bias caused by low transcript numbers. Hence, samples with a low bacterial load were less likely to be included in the analysis. Additionally, several samples were lost by the clinical microbiology lab and unavailable for culture and RNA extraction. Thus, only 6 of 18 *in vivo* samples were included in the transcriptomic analysis. Bias in the differential gene expression analysis introduced by these factors cannot be excluded.

## Conclusion

In summary, we found the gene expression profiles of *Haemophilus influenzae* to differ widely between lab-cultured and clinical isolates. *In vitro* isolates conformed with regard to their mRNA transcription profiles, while the gene expression of *in vivo* bacteria was considerably more diverse. The differential gene expression patterns we found were less distinct compared with previous studies that used *in vitro* isolates or controlled murine protocols. Undoubtedly, this is due to the inherent transcriptional heterogeneity of *in vivo*-growing bacteria as a consequence of uncontrolled and differing growth environments in human airways. Genes that are involved in major metabolic pathways were generally expressed at lower levels *in vivo,* while those involved in the synthesis of essential components of nucleic acid, molybdopterin acquisition enzymes, and transferrin-binding proteins were upregulated in this condition. The generated data set is suitable for hypothesis generation in future studies, focusing for instance on the stress response gene *msrAB* and the highly expressed hypothetical proteins HI_1599 and HI_1600. It is also suitable as a reference against which other *in vitro*-generated data sets can be compared to determine how well the used laboratory conditions used mimic the *in vivo* environment. These results also raise questions about the generalizability of microbiological studies that are performed under controlled circumstances, which are unlikely to approximate the diverse environments encountered by pathogenic bacteria in human airways. As such, the discoveries made in our material may be particularly significant, since they implicate processes that are ubiquitous for the survival and pathogenic ability of *H. influenzae* isolates *in vivo*.

## ACKNOWLEDGMENTS

We would like to thank the students and staff at Skåne University Hospital for their help with patient enrollment and sample collection, and Dr. Eleni Touriki at the Clinical Microbiology Department in Lund, Sweden, for sample processing and storage. We would also like to acknowledge Prof. Marvin Whitely and Dr. Dan Cornforth at the Georgia Institute of Technology and Prof. Thomas Bjarnsholt and Dr. Blaine Fritz at the University of Copenhagen for introducing us to RNA extraction methods and bacterial transcriptomics.

Support by NBIS is gratefully acknowledged. The computations were performed on resources provided by SNIC through UPPMAX under Project SNIC sens2020512 and LUNARC at Lund University. The authors would like to acknowledge Clinical Genomics Lund, SciLifeLab, and the Center for Translational Genomics, Lund University, for providing sequencing and analysis expertise and service. This publication used the PubMLST website (https://pubmlst.org/), developed by Keith Jolley (84). The

development of this website was funded by the Wellcome Trust. We thank AdvanSci Research Solutions (https://www.advansci-research.com) for editing an English draft of this manuscript.

This study was financed by Swedish Governmental Funding of Clinical Research (ALF), Swedish Research Council 2018-06924 and 2021-06380, Swedish Society for Medical Research (SSMF), the Royal Physiographical Society, Skåne County Council's Research and Development Foundation, and the Gyllenstiernska Krapperup Foundation. The funders had no role in study design, data collection and interpretation, or the decision to submit the work for publication.

## AUTHOR AFFILIATIONS

[1]Infection Medicine, Department of Clinical Sciences Lund, Medical Faculty, Lund University, Lund, Sweden

[2]Clinical Microbiology, Office for Medical Services, Region Skåne, Lund, Sweden

[3]Experimental Infection Medicine, Department of Translational Medicine, Medical Faculty, Lund, Sweden

## AUTHOR ORCIDs

Linnea Polland http://orcid.org/0000-0003-1912-265X
Yi Su http://orcid.org/0009-0001-4802-5210
Magnus Paulsson http://orcid.org/0000-0003-1104-2727

## FUNDING

| Funder | Grant(s) | Author(s) |
| --- | --- | --- |
| Vetenskapsrådet (VR) | 2018-06924, 2021-06380 | Magnus Paulsson |
| Svenska Sällskapet för Medicinsk Forskning (SSMF) | | Magnus Paulsson |
| Gyllenstiernska Krapperupsstiftelsen (Krapperups-slott) | | Magnus Paulsson |
| Kungliga Fysiografiska Sällskapet i Lund (Royal Physiographic Society in Lund) | | Linnea Polland |
| | | Hanna Rydén |
| | | Magnus Paulsson |
| Skåne County Council's Research and Development Foundation (Skane County Council's Research and Development Foundation) | | Magnus Paulsson |

## AUTHOR CONTRIBUTIONS

Linnea Polland, Data curation, Formal analysis, Investigation, Validation, Visualization, Writing – original draft | Hanna Rydén, Formal analysis, Investigation, Writing – review and editing | Yi Su, Data curation, Software, Writing – review and editing | Magnus Paulsson, Conceptualization, Data curation, Formal analysis, Funding acquisition, Investigation, Methodology, Project administration, Resources, Supervision, Writing – review and editing

## DATA AVAILABILITY

The FASTQ files from RNA-sequencing that have been depleted of human reads are available from the European Nucleotide Archive under study accession number PRJEB60515. Assembled genomes of the clinical isolates are available as FASTA sequences at PubMLST. The R code used in this study can be found in Supplemental File 1. The command lines that were used for the pan/core genome creation, processing of

sample reads, and pseudoalignment can be found in Supplemental File 2. The annotated core and pan genomes are available upon request.

## ADDITIONAL FILES

The following material is available online.

### Supplemental Material

**Supplemental file 1: R scripts (Spectrum01639-23-s0001.docx).** R scripts used for differential gene expression analysis and figures

**Supplemental Tables S1-5 and Figure S1-3 (Spectrum01639-23-s0002.docx).** Supplemental tables about genomes used for pan genome creation, clinical data about study subjects, basic sequencing data and GO-term analysis. Supplemental figures about the pan genome creation and differential gene expression using the pan genome as reference.

**Supplemental file 2: Bioinformatics commands (Spectrum01639-23-s0003.docx).** Code and commands used for bioinformatic analysis

**Supplemental file 3: Taxonomic data (Spectrum01639-23-s0004.xlsx).** Taxonomic analysis from RNA-seq of all clinical samples

**Supplemental file 4: Read counts (Spectrum01639-23-s0005.xlsx).** Read count data using core and pan genomes as reference

**Supplemental file 5: DESeq2 result core genome (Spectrum01639-23-s0006.xlsx).** DESeq2 result file with logarithmic fold change of all genes in the core genome

**Supplemental file 6: DESeq2 result pan genome (Spectrum01639-23-s0007.xlsx).** DESeq2 result file with logarithmic fold change of all genes in the pan genome

**Supplemental file 7: DESeq2 result DEGs only (Spectrum01639-23-s0008.xlsx).** DESeq2 result file with logarithmic fold change of significantly changed genes in the core genome

**Supplemental file 8: GO annotation (Spectrum01639-23-s0009.xlsx).** Gene ontology annotation of genes that were significantly changed (DEGs) and not significantly changed (Non-DEGs)

### Open Peer Review

**PEER REVIEW HISTORY (review-history.pdf).** An accounting of the reviewer comments and feedback.

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
