## [Reviewer comments · Microbiology Spectrum]

Microbiology Spectrum

In vivo gene expression profile of *Haemophilus influenzae* during human pneumonia

Linnea Polland, Hanna Rydén, Yi Su, and Magnus Paulsson

Corresponding Author(s): Magnus Paulsson, Lunds Universitet

Review Timeline:

Submission Date:	April 19, 2023
Editorial Decision:	May 23, 2023
Revision Received:	July 10, 2023
Accepted:	July 12, 2023

Editor: John Atack

Reviewer(s): Disclosure of reviewer identity is with reference to reviewer comments included in decision letter(s). The following individuals involved in review of your submission have agreed to reveal their identity: Jeroen Daniel Langereis (Reviewer #3)

Transaction Report:

DOI: <https://doi.org/10.1128/spectrum.01639-23>

May 23, 2023

Dr. Magnus Paulsson
Lunds Universitet
Infection medicine, Department of Clinical Sciences Lund, Medical faculty
Lund
Sweden

Re: Spectrum01639-23 (The Haemophilus influenzae in vivo gene expression reveals major clues about bacterial central metabolism, acquisition of trace elements, and other essential pathways during infection of the human lung)

Dear Dr. Magnus Paulsson:

Link Not Available

Sincerely,

John Atack

Journals Department
Reviewer comments:

Reviewer #1 (Comments for the Author):

Polland et al. report the in vivo transcriptome of Haemophilus influenzae during pneumonia in human patients. Elucidating gene transcription in vivo in specific clinical niches is of high interest and offers opportunities to identify important pathogenic processes and potentially new interventions for disease states. As such the overall objective of this manuscript is beneficial to the research community. However, there are several issues with the manuscript in its current state.

Specific Comments:

1, The choice of a group level comparison is problematic, and the authors acknowledge in their discussion that pairwise comparison may offer "further interesting discoveries". This problem is compounded by the choice of strains - it is not clear why

the specific clinical strains were chosen. Eighteen patients had *H. influenzae* cultured from their lungs. Of these 18, ten samples were excluded due to low levels or poor-quality mRNA or contamination with other *Haemophili*. Additionally, 12 isolates were unavailable for *in vitro* analysis, although this is unexplained and seems odd given that the bacteria were identified by culture. Figure 1 then shows nine study subjects being analyzed - five both *in vivo* and *in vitro*, three only *in vivo*, and one only *in vitro*. So only five samples had directly paired *in vivo* and *in vitro* derived mRNA, it would have been better to focus on the directly comparable samples. Additionally, the type strain included in the *in vitro* analysis seems an odd choice since it was originally isolated from a 10-year old patient with acute otitis media. Why not choose a strain isolated from an adult with lung disease like the patient cohort? It is not clear what benefit inclusion of any type strain in the *in vitro* analysis provides.

2, *In vitro* growth. It is stated that bacteria were grown in complete media and sampled at 240 minutes or in the exponential phase. Where in the exponential phase? Were all isolates growing at the same rate? It would be better to give a specific OD at the point of harvest. Also, the bacteria were pelleted by centrifugation and then RNA Later added. Centrifugation can cause transcriptomic changes. It might be better to add culture directly to RNA Later rather than centrifuge the samples.

3, The volcano plots (Figure 3 and Figure S3) raise a question. Both have a spot labeled HI_1600, however they are in different locations. Should they not be located in the same position? Or are they derived from different data sets. Figure 3 has a spot labeled *tbp1* and S3 a spot labeled *tbpA*. First the correct gene name is *tbpA* (the designation *Tbp1* is correctly used only for the encoded protein), nevertheless the *tbp1* spot in figure 3 is not in the same location as the *tbpA* spot in S3.

4, The discussion is overlong and too speculative.

Minor points:

1, Line 37. *tbp1* should be *tbpA*, and additionally *fbpA* is not a transferrin acquisition gene rather it is an iron acquisition gene. The incorrect use of *tbp1* for the gene occurs often in the manuscript.

2, Lines 69-70, " " while NTHi is a more heterogeneous group that are subdivided ..." should be "... while the NTHi are a"

3, Line 430. Overturned should be turned over. Overturned does not mean the same thing as turned over.

Reviewer #2 (Comments for the Author):

Review of "The *Haemophilus influenzae* *in vivo* gene expression reveals major clues about bacteria central metabolism, acquisition of trace elements, and other essential pathways during infection of the human lung"

By Linnea Polland, Yi Su, and Magnus Paulsson

Summary: The authors carried out the first global transcriptome analysis of *Haemophilus influenzae* (Hflu) bacteria directly from human lung infections. Sputum or bronchial wash specimens from pneumonia patients were screened by standard clinical micro culture for Hflu, and where found, an isolate was collected for *in vitro* RNA-seq, and total RNA from the banked clinical specimen was processed for *in vivo* RNA-seq. In the end, the authors produced core gene transcriptome profiles of Hflu cells living within 9 cases of pneumonia infection, and they compared these to *in vitro* transcriptome profiles from 9 matched clinical isolates in exponential phase in rich medium. The *in vitro* grown cells all had relatively comparable core gene transcript profiles, whereas *in vivo* transcript profiles were distinct from the *in vitro* samples and had higher within-group variance. A DESeq2 analysis identified genes that were differentially expression in *in vivo* samples compared to *in vitro*. Specifically, differentially expressed transcripts were identified with: (a) reduced relative expression encoded proteins involved in energy production, and (b) increased relative expression encoded proteins involved in stress responses and nutrient acquisition.

General assessment:

This is the first analysis I've seen to measure global gene expression profiles of *Haemophilus influenzae* directly from human infections, and I believe this is an important dataset to release. The authors have put in a large effort and expended many resources to acquire this dataset, and their analysis provides insights into bacterial gene expression within a human infection in contrast to typical *in vitro* growth conditions.

However, I have three large issues with the manuscript as is: (1) the MS reads a bit like a draft and needs substantial editing for grammar, style, clarity, and content; (2) the genomic and bioinformatic methods and QC steps are poorly described, so some of the major elements of their approach are obfuscated; and (3) the analysis emphasizes how the expression profiles in human infections differ from 'standard *in vitro* conditions' but they don't contextualize this 'standard' condition as log-phase shaking culture in rich media in their discussion. This context makes some of their findings a little more straightforward, especially if compared to some other Hflu transcriptional regulation, RNA-seq/microarray, and Tn-Seq experiments that have been published. Addressing these issues would improve the MS considerably without necessarily carrying out new or different analyses.

Main comments:

This MS needs a strong English language edit. In addition, the motivation and purpose of many parts of the analysis are not provided, and the description of the analyses themselves can be vague, even with the supplementary files. Including stronger reasoning for the analysis approach would help make this work more digestible. It is possible that the authors need to get more

detailed information or consult with whomever helped them with the initial data processing steps to go from reads to count tables (UPPMAX?)

More description and results associated with the in vivo experiment are needed, especially related to the fact that most sequenced RNA molecules were not from Hflu. This is not a surprise but since data like this have not been reported before, it is important to discuss the results that lead to the Hflu transcript profiles. The Supplement Table S3 needs to include total reads mapped to their "pan-transcriptome" reference for Hflu. There appears to be a very wide range of read counts per transcriptome, so this needs to be reported. The extremely large difference in useable read counts between in vivo and in vitro is also notable, and it may affect aspects of the analysis. It would also be useful to report on how well the rRNA depletions used worked.

Descriptions of the bioinformatics approach were convoluted by dividing up information between main and supplement, but then these together also did not adequately explain how the raw data were processed into count tables. I could infer what was done, but this raised complex issues. I don't think all the analysis issues necessarily need to be fixed but the authors should acknowledge the strengths and limitations of their bioinformatics approach:

- I understand why a pan-genome reference was used, and how Roary output was used to build a "pan-transcript" reference for Kallisto, how Kallisto was used to count, and how the initial count tables were imported for DESeq2 analysis, etc. However, none of this is described.
- The authors need to acknowledge and consider the types of read counting artifacts that can arise from their approach. This is especially true because of how they do their Kallisto-based counting.
- What is the purpose of the control 3655 strain? It's not even in the pan-transcript reference to check how this approach might be affected by sequence divergence among strains.
- Are there genome assemblies of the clinical isolates? This appears to be stated in the supplement, but I don't think this is what the authors mean? It would obviously be much better if their pan-transcriptome was made from annotated assemblies of the actual strains they used in their experiment. They know the MLSTs and there are hundreds of Hflu genomes in public databases, so they might've been able to match up their reads to better references. Why were the 15 used chosen? Why wasn't 3655 included? If they'd used many more, they could have used an even more reduced core gene list.
- Even without redoing any analyses, it would be worth discussing or commenting on the strengths and limitations (especially with respect to accurately counting reads) of their approach.
- Which core genes were used to do the GO analysis? The GO analysis sections also need heavy editing for clarity and meaning.
- How were analyses done when including all genes? Were missing genes treated as zeros or NAs?

Detailed comments:

Lines 2-4: Cumbersome grammar and very long for a title. Maybe something like: "In vivo gene expression profiles of *Haemophilus influenzae* during human lung infection reveal..."

21-43: Expand to explain more about slightly more about data acquisition and analysis pipeline, since this is non-standard due to diverse strains. E.g. what fraction of sequence reads were bacterial in the in vivo experiment?

47: in vitro and animal studies. Worth emphasizing here and elsewhere that animal models of *H. influenzae* lung infection are relatively poor.

47: "Avoid"? Or to understand the biases? How would these results compare to "dual RNAseq" of NTHi+host cells?

50-51: How is this quantifiable from the PCA?

52-55: How does this relate to how "vital" the pathways are?

62: "evolutionally developed" to "evolved as a"

63-71: Genomic details are of some interest, but this is awkward. How does this fit with the aims here? Given that most of the genomic heterogeneity is ignored, this introduction might better focus more on the inadequacy of in vitro and animal models, though it's worth mentioning the high diversity. Unclear a discussion of capsule is really warranted much here.

72-83: The Pa example is fine; probably worth citing some others that also show this sort of thing. This could also use a discussion of animal models of NTHi lung infections, for which there are some. Also some literature on bacterial-host cell that might be of interest, or at least further expand in the discussion.

97-113: Should this refer to Supplementary Table S2?

119: taxonomic analysis of genome assemblies using Kraken2?

121: Why was this reference chosen?

125: Were all strains in exponential phase after 4 hours? Approximate ODs? Growth curves may be useful here. Choice to use duplicates should be specified.

133: These supplementary methods might as well be added to the main MS.

133: Supplementary Table S3 is incomplete. What is the total number of Hflv reads mapped for each sample? What are the public accession numbers of

136-144: Is this section out of place? Supplement Tables S2 and S3 belong before. Table S3 is incomplete.

148-150: Might as well move the missing steps out of supplement and into the main, or at least give a short version. It is key to specify

Supplement File S1 Bioinformatic methods: This is a bit of a mess. May the authors need help from UPPMAX? It implies there are also genome assemblies of the individual isolates? If so, these methods and datasets also need to include explanations of the genome assembly process. If a genome assembly was performed, why weren't the seven strain isolates included in the pangenome use with Kraken for the gene counting? Or was there no genome sequences, and these are just using the pan-genome calculations with the reference strains to count these RNA seq reads with Kallisto? Or were there transcript assemblies? There are a lot of missing moving parts here that strongly affect interpretations, etc.

150-153: This needs to explain what was done with non-core genes.

163: All core genes, or representative core genes from a specific reference genome?

171-172: Fine if no multiple testing correction but should specify p-cutoff.

175-177: Supplement Table S3: This is inadequate. Please include total H. influenzae reads mapped. Please include individual accessions to samples. Please include final count table. Please explain why some files have R1 and others R1 and R2? How was the difference handled in the analysis? It would be desirable to include the count table as a supplement if possible, or deposit and EMBL or NCBI as a gene expression dataset.

184-187: This might refer to Supplementary Table S3. What are the QC criteria? What is the mean and range of read counts from the in vivo samples? I believe that there's likely enough Hflv reads here for these comparisons, but the read counts must be relatively low, especially compared to the in vitro samples. This might affect some aspects of downstream analysis (e.g. comparing transcript profiles from 200K reads to 20M reads). Importantly, % of Hflv reads that are from rRNA would be also helpful to know. This gave specifics for all read files, but it would also be useful to have these stats aggregated by sample.

188-191: The exact condition needs to be specified, that these are cells growing in rich media and are in exponential phase. The dramatic differences between these and the in vivo profiles are quite interesting, but other in vitro conditions may look more similar, for example in a nutrient deprived culture.

200-206: Explain the logic. I think I understand, but the implicit use of this reference with Kallisto for the RNAseq experiment suggests reasonable logic. There's a lack of clarity about whether the cultured isolates also had genome sequences and why these weren't included. If these are all NTHi, was it important to include typeable strains? Why only these 15 strains? Wouldn't knowing the MLST at least help pick appropriate references? Was Roary used with all default settings? Why isn't NTHi3655 included in the study?

206-212: This is too packed up and needs to be detailed. How are reads counted against the pan-genome? Are only core genes included in the analysis? Could there be any issues mapping reads from some of the isolates to the pan-genome that might affect the analysis? It would be worth acknowledging these issues.

211-214: This is all very interesting but is stated without interpretation. Worth stating that this contrast is against exponentially growing cells in nutrient rich conditions. An outcome would also be worth stating. It probably makes more sense to present the PCA first, since the DEG analysis depends on this to a large extent. Also as a note of caution, one assumption of DESeq2's default normalization is that most genes aren't differentially expressed. Did the DEG analysis consider using clinical source (subject) as a co-variate in an LRT to help account for the heterogeneity among the bacterial genomes?

217: The PCA might belong prior to DEGs, since it clarifies how the samples do/do not cluster by genotype and condition.

224: Is the heatmaps only of high-variance, or does it use the DEG analysis? Clarify.

225-227: This grammar implies that the clinical isolates are distinct from the lab-grown ones, when in fact, the authors are comparing Hflv transcript counts from lung samples versus from in vitro log-phase samples using a strain isolated from the

same lung samples. This is very different. What's meant is that the in vivo samples showed higher heterogeneity than diverse strains grown in a single condition. This shows that a lot of the heterogeneity is probably driven by how variable the conditions are, rather than the genomic heterogeneity among the isolates grown in broth.

227-228: How are accessory genes treated in the pan-genome heatmap? As zeros? Why would the clustering differ, and what might drive these differences?

229-233: This is a confusing presentation. How was the analysis conducted? Were gene absences treated as NAs? As zeros? If the latter, this would cause a lot of oddities. The focus on core is reasonable, but also there are so many artifacts that could be arising during the Kallisto-based mapping.

235: Organizationally, still makes sense to bring DEGs down here, since this next section is about those genes.

237: Stability within-group was not shown. Perhaps plotting individual genes between conditions would clarify. I suspect the variance within the in vivo samples will generally be much higher than within the in vitro samples, even if there are still statistically significant differences.

242: Higher relative expression.

244: What is meant by "interaction"? Correlated expression with *msrAB*, or some other unspecified link? What is meant by "connected" in this context?

246-247: Strong evidence support a role for these genes in transformation but also their regulation is known to depend on depletion of preferred sugars and purine precursors.

251-252: For all DEGs or from all genes? I think this must have used some specific reference genome and reduced to only the core protein-coding genes? What fraction of genes got GO annotations?

255: Please comment on genes of unknown function or genes not annotated by Panther.

260-263: Grammar is convoluted. These processes were not "less common".

263-264: This is a good follow-up. What is the obvious interpretation?

264-267: Compared to shaking culture that should be well oxygenated?

272: "Less common" is strange grammar. The fraction 2 of 11 means something different than implied here.

277-285: Grammar and interpretation are still complicated here. Given that several of the genes discussed above are in these processes, connecting the "most different" DEGs and these pathway analyses would probably help.

286: No measurements of Mo-transport were performed. Clarify.

298-299: Excellent goal. Worth specifying that the condition chosen was rich-media shaking (well oxygenated) during exponential phase.

299-301: Thank you. This is great!

304-305: This is primarily due to the default paralog splitting done by Roary. Homologs merged would give a smaller cloud. This is minimized by use of complete genomes. Why were those the choices, and what is the relevance of that pan-genome with respect to the goals of the paper? The use of this for mapping seems like it was the main purpose, and for defining a core genome. But those aren't really discussed. Using a Kallisto-based approach was an interesting, potentially cool decision but has some implications and potential counting artifacts that aren't discussed. For example, even the reference genome wasn't included in the pan-genome reference, so the ability to even detect counting artifacts wasn't really feasible.

305-308: This assumption is probably true, but no evidence is presented for this.

310-312: Well, that study was using a single strain, so included all its genes. Although the proportion is distinct, the power was also probably quite different in those lab experiments.

315-318: These are fantastic observations, but I might re-word or re-think this. I would argue that only the in vitro grown cells show strong clustering. Sure, the in vivos are more to one side of the in vivo cluster in the PCA, so there's nice strong differential expression, but there's still clearly a great deal of heterogeneity here in the in vivo samples. This could be because of many reasons: e.g. much lower sampling of reads in the Hflu transcriptome, no opportunity for replication, not sure the infections

were monoclonal, unknown genotypeXenvironment interactions that are irrelevant in log-phase broth culture, etc. These would be interesting to discuss.

331-338: These are great speculations, and potentially biofilms are important, but also other types of nutritional shifts, like simply stationary phase, lower oxygen, etc could be important. In particular, I would guess that CRP response is on, and relative induction of purine biosynthesis and competence genes suggests also suggest a nutrient limited state.

342-343: Wasn't the RNA from liquid cultures? Were the shaking cultures also in high CO₂ or is this just how the plates are grown?

350-352: It's a smaller fraction used for recombination but regardless, the state of competence is induced by specific nutritional conditions. Might be worth noting that the competence genes have been shown to be involved in biofilm formation.

363: Another competence connection.

373-375: Worth citing/discussing in vivo mouse lung Tn-Seq experiments + and - flu infection, as these also hit purine biosynthesis and some others seen here, so provides a functional connection.

376-391: Good discussion but might also bring up relevant old microarray experiments that find this and related iron acquisition genes differentially regulated in various in vivo conditions. I am also a little lost about directionality. Which setting appears to have more access to oxygen, based on the gene expression analysis?

376-403: It is important to remember that all of this is in contrast to growth in rich media under non-nutrient limiting conditions. It doesn't seem surprising that either iron or Mo are going to be way more limited in lungs than in rich media at log-phase.

404-410: This seems unneeded? Both of the next paragraphs are about redox, which related very tightly with previous discussion of TCA, iron, and oxygen.

411-428: Cool discussion. Neat findings.

429-441: This could use expansion about the issues of doing this, besides RNA quality. That's fine. There's no poly-A selection here so that should be a major problem. But the massive amount of human RNA is definitely a problem for getting this type of work done, and the few useable reads that emerge present some difficulties of interpretation. What about the bioinformatics problems associated with use of the reference pan-transcriptome? Allelic variation is also very high among Hflu strains, so read mapping (be it actual or quasi-alignment) may undercount for some alleles versus others, especially since the strains used in the experiment weren't the strains used in construction of the reference pan-genome.

429-441: What about the possibility of polyclonality in infected samples?

455+: I think a key outcome is that we have a better idea of the environment experienced by Hflu when in the lung, and it is possible that comparing this data to other data in other in vitro settings would look closer to these real samples, so that we can better mimic in vivo experiments.

Reviewer #3 (Comments for the Author):

I have read the manuscript with great interest, and I think this is a good example to show that in vivo tests are not always translatable to the in vivo situation.

The study setup is clear and even though there might be a bias because of the large number of samples that did not meet the quality controls, the in vivo transcriptional data is valuable information.

I'm not familiar with the current rules regarding data sharing, but I would encourage the authors to also share their R codes to enable replication of the data and control on the data analysis pipeline.

In this discussion line 440 is mentioned that samples with less than 1% transcript assigned to *H. influenzae* were excluded. Could the authors include a supplemental table of the percentage *H. influenzae* transcripts in comparison to total transcripts, and also a list of other pathogens with percentages found within the transcripts? This could give some inside which co-infections were possible present.

I would prefer to include supplemental file 1 into the material and methods section of the main manuscript.

Although the discussion is already extensive (but very useful), I would prefer to include the discussion of supplemental file 5 into the main manuscript.

There are track changes present in supplemental file 1.

Supplemental file 5 contains some minor textual error, such as a missing space between *H.influenzae* in the first and third sentence.

Reviewer #4 (Comments for the Author):

This manuscript by Polland and collaborators compares the transcriptome of *Haemophilus influenzae* in vivo with that in vitro. Although a large number of patients were initially included, only a few samples were ultimately analyzed. Therefore, the work does not provide sufficient evidence to draw a general conclusion regarding the metabolic status of NTHi in vivo. This is due to the limited number of samples (n=9) tested in vivo, which had varying patient statuses, as well as the small number of reference genomes (n=15) used to create the limited core genome that formed the basis of the DEG analysis. However, the PCA plots in Figure 2 present an alternative view. Unfortunately, the data novelty related to in vivo infection is quite limited when compared to previously published models (such as those involving animal/primary cells and cell lines).

1. Line 21. *Haemophilus influenzae* is not commonly associated with HAP. This should be corrected in the Introduction.
2. The authors should acknowledge previous studies on *H. influenzae* transcriptomics in animal models.
3. Line 79. *Pseudomonas aeruginosa* does not belong to the same species and has a considerably larger genome than *H. influenzae*.
4. Line 84: It is unclear what "complicated" refers to, and the authors should provide more context or clarification.
5. Line 86: The *H. influenzae* in vivo transcriptome is not unknown (Aziz is cited in the Discussion and should also be cited in the Introduction.
<https://www.frontiersin.org/articles/10.3389/fcimb.2021.723481/full>
<https://doi.org/10.1016/j.csbj.2021.05.026>
<https://www.frontiersin.org/articles/10.3389/fmicb.2019.01622/full>
6. Line 88: Several studies have shown a significant discrepancy between in vivo and in vitro conditions; thus, it is a fact rather than a hypothesis.
7. Line 180, and Table 1 and 2: The sample description is unclear and confusing, particularly when attempting to relate it to Figure 1. Therefore, the figure legend for Figure 1 could be made more descriptive and clear.
8. Line 187: It is unclear what is meant by "picking colonies" from routine cultures from only 6 patients when samples from 8 patients were analyzed for transcriptomics. It is confusing why an equal number of samples were not chosen for downstream analyses. Additionally, why were only two colonies picked, and did these colonies undergo WGS?
9. Line 189: Strain "3655" is not described in the manuscript. It is unclear why it was included in the study since it is not sequenced according to Supplementary Table S1.
10. Line 192: Although the manuscript states that eight patients were analyzed, Table 1 shows nine patients in the "study cohort," which is confusing.
11. In general, a cohort is defined as "a group of persons, usually 100 or more in size".
12. Line 202: Why were capsulated strains (small in number) included when you were searching for the core genome with a high number of NTHi strains? I think you should only focus on NTHi, excluding the capsulated strains, especially when you did not have a good number of capsulated strains (only 3); the sample pool is rather weak here. The information on core genomes of NTHi (various clinical isolates and reference genome, more than 12 strains) has been widely published in several studies and should be referred to in your study.
13. Line 211-214: It is unclear what the difference is between the 328 genes of the core genome (defined as DEG) vs. the total core genes (1067). "In total" of what? Does this refer to the 9 isolates and 3655 reference genome (line 205)?
14. Figure 2 is confusing regarding the clarification of in vitro isolates (line 187). I assume those with "_1 and _2" are 2 colonies picked from 6 patients for in vitro analysis. The authors need to make the indication clearer in the figure legend. Why did the in vitro sample of HI_LUIN_26_1 and _2, even when in in vitro grown conditions, not cluster together and instead showed a very discrepant heat map profile, despite being from the same isolate, compared to other _1 and _2 isolate pairs?
15. Line 225, "in vivo samples of clinical isolates" instead.

16. Line 227. Figures 2C and D are difficult to comprehend and should be simplified by using different color labeling instead of both colors and names for different strains. The same applies to Figure 4, where the use of different colors for strains and conditions can be confusing for the reader.
17. Figure 4: LUIN_28 is missing.
18. Line 268: Figure 5 can be removed or moved to the Supplementary data. While it is useful to provide some background information, this information can easily be found elsewhere. Instead, it would be more informative to state that all transcripts were downregulated compared to specific reference genes.
19. Figure 6: *pfkA* is down-regulated in vivo. The mRNA level is very low compared to other transcripts, and it is unclear whether this transcription is significant. The authors should provide information on the threshold (cut-off) used in their analyses? The same applies for *napA*, which has a p-value below 0.01.
20. Figure 6: Only 6 transcripts show a significant change when comparing in vitro and in vivo "expression" are compared. This finding contradicts Figure 5, which suggests that all gene products encoding enzymes shown in red and green colors are changed between in vitro and in vivo conditions. The authors should address this discrepancy and clarify the results.
21. Figure 7A: *AspA* is found to be significantly different between in vitro and in vivo conditions, but it is not shown in Figure 5. The authors should explain why *AspA* was not included in Figure 5 and provide a clear rationale for its inclusion in Figure 7A.
22. Line 272: The authors state that "Iron transport and intracellular sequestering (2/11) were less common in vivo compared to in vitro (Figure 7B)." This statement is not accurate as the Authors only studied transcripts. Unless phenotypic data are available but not shown in the study, this statement appears to lack supporting evidence.
23. Line 277: Similar to the comment above, the authors state that purine and pyrimidine metabolisms were changed. However, this statement is not accurate as the authors only analyzed transcripts. It would be more appropriate to state that transcripts involved in purine and pyrimidine metabolisms were changed.
24. Line 286: The authors state that molybdate-transport was changed, but it is unclear whether this was tested. It would be helpful to clarify whether the statement is based on the transcript analysis alone or if there is additional evidence supporting this claim.
25. Line 292: The authors state that many enzymes involved in a particular pathway were less expressed. It would be useful to indicate whether the authors checked the expression levels of these enzymes using western blots or enzymatic assays, and if not, to modify the statement to reflect that only transcript levels were analyzed.
26. Discussion is extensive long and needs to be shortened.
27. Line 315-317: it is not surprising that bacteria are clustered based on growth conditions, despite diversity in genetic background, as core genes (housekeeping genes, Line 307) were targeted for DEG analysis.
28. Line 332-333: the stage of diseases severity (status of inflammatory and immune response) might need to be considered here, as it may also affect the growth fitness or dormancy of bacteria during in vivo. Authors had mentioned the similar impact in Line 339-342.
29. From my point of view, it is not strange that mRNA levels are different in vivo as compared to in vitro using excess NAD and hemin in the very rich BHI broth. The authors do not comment upon this.

Reviewer #5 (Comments for the Author):

The manuscript Spectrum01639-23 by Polland et al presents an interesting approach to study the in vivo *H. influenzae* transcriptome in comparison to the in vitro obtained transcriptome, providing a good starting point for in-depth real-life approximation.

Because in vitro pure bacterial cultures are a more controlled and simple approach, researchers frequently overlook the fact that the culture conditions do not correspond to those encountered in patients. Although results from in vivo models are always more valuable, care must be taken when analyzing the data because this is a non-controlled environment with many external confusion factors that may affect the results.

The manuscript is clear and well-written. The authors clearly stated their objectives and used the appropriate methodology to achieve the results. The manuscript is well-structured, and the results are clear. The supplementary data adds value to this work by reporting and discussing in detail some additional results that did not fit into the main manuscript document. The work has been well planned, with an emphasis on avoiding or reducing external and methodological factors, however, I have some concerns regarding the use of the *H. influenzae* clinical strains.

My first doubt regards the indistinctive use of non-typeable *H. influenzae* (NTHi) and capsulated strains; there are clear differences between both types of strains, but NTHi are the most commonly identified in respiratory infections. Besides, only some serotypes were included (B, D and F). I did not see in the description of the nine studied strains whether they are all NTHi or if there are any capsulated strains (this information, along with MLST information, should be included in Table 2). If all of the test strains are NTHi, consider making the core and pan genome using only NTHi strains.

Second, after reading the entire manuscript I have some doubts about the origin of the *H. influenzae* used in vitro. Theoretically, the strains used for in vitro testing should come from the samples studied in vivo; however, the methodology explaining the origin of these strains is ambiguous, and the results (Figure 2, color legends) show that there is only coincidence in five of the

nine strains (LUIN_26; LUIN_33; MAAK_03; MAIV_24; MAIV_32). On the other hand, LUIN_28 was only studied in vitro, while LUIN_28, LUIN_29, LUIN_31 and MAIV_34 were only studied in vivo. Despite that, in lines 219 to 221 say "... the in vitro cultured bacterial isolates cluster closely together regardless of genetic background. The in vivo gene expression showed a higher transcriptional diversity and did not cluster with the corresponding bacterial isolated cultured in vitro". This sentence implies that gene expression was studied in vivo and in vitro on the same strains. This should be revised because, in order to draw any conclusions, the strains used in vitro should come from their original clinical samples in order to reduce differences. It is obvious that clinical samples may contain more than one H. influenzae lineage (this cannot be controlled experimentally) but the ones isolated and used for experimental testing should be directly linked to the in vivo testing.

Table 2 shows the presence of two different H. influenzae lineages (ST-12 and a new MLST) in sample LUIN_26. Which strain was used in the in vitro tests? Or were they both used? The ST numbers of the newly identified MLSTs should be included (ask for the ST number on the MLST page).

Minor comments

Lines 201-202: "... including genomes from NTHi serotype A, B, D and F..." In Supplementary Table S1, describing the strains used for Core and Pan genome determination, there is no serotype A strain.

Lines 201-202: "... including genomes from NTHi serotype A, B, D and F..." This sentence brings confusion because it seems that NTHi have serotypes, use instead "... including genomes from NTHi and serotypes b, d and f".

Staff Comments:

Preparing Revision Guidelines

Please return the manuscript within 60 days; if you cannot complete the modification within this time period, please contact me. If you do not wish to modify the manuscript and prefer to submit it to another journal, please notify me of your decision immediately so that the manuscript may be formally withdrawn from consideration by Microbiology Spectrum.

Polland et al. report the *in vivo* transcriptome of *Haemophilus influenzae* during pneumonia in human patients. Elucidating gene transcription *in vivo* in specific clinical niches is of high interest and offers opportunities to identify important pathogenic processes and potentially new interventions for disease states. As such the overall objective of this manuscript is beneficial to the research community. However, there are several issues with the manuscript in its current state.

Specific Comments:

1, The choice of a group level comparison is problematic, and the authors acknowledge in their discussion that pairwise comparison may offer “further interesting discoveries”. This problem is compounded by the choice of strains – it is not clear why the specific clinical strains were chosen. Eighteen patients had *H. influenzae* cultured from their lungs. Of these 18, ten samples were excluded due to low levels or poor-quality mRNA or contamination with other Haemophili. Additionally, 12 isolates were unavailable for *in vitro* analysis, although this is unexplained and seems odd given that the bacteria were identified by culture. Figure 1 then shows nine study subjects being analyzed – five both *in vivo* and *in vitro*, three only *in vivo*, and one only *in vitro*. So only five samples had directly paired *in vivo* and *in vitro* derived mRNA, it would have been better to focus on the directly comparable samples. Additionally, the type strain included in the *in vitro* analysis seems an odd choice since it was originally isolated from a 10-year old patient with acute otitis media. Why not choose a strain isolated from an adult with lung disease like the patient cohort? It is not clear what benefit inclusion of any type strain in the *in vitro* analysis provides.

2, *In vitro* growth. It is stated that bacteria were grown in complete media and sampled at 240 minutes or in the exponential phase. Where in the exponential phase? Were all isolates growing at the same rate? It would be better to give a specific OD at the point of harvest. Also, the bacteria were pelleted by centrifugation and then RNA Later added. Centrifugation can cause transcriptomic changes. It might be better to add culture directly to RNA Later rather than centrifuge the samples.

3, The volcano plots (Figure 3 and Figure S3) raise a question. Both have a spot labeled HI_1600, however they are in different locations. Should they not be located in the same position? Or are they derived from different data sets. Figure 3 has a spot labeled *tbp1* and S3 a spot labeled *tbpA*. First the correct gene name is *tbpA* (the designation *Tbp1* is correctly used only for the encoded protein), nevertheless the *tbp1* spot in figure 3 is not in the same location as the *tbpA* spot in S3.

4, The discussion is overlong and too speculative.

Minor points:

1, Line 37. *tbp1* should be *tbpA*, and additionally *fbpA* is not a transferrin acquisition gene rather it is an iron acquisition gene. The incorrect use of *tbp1* for the gene occurs often in the manuscript.

2, Lines 69-70, “ ” while NTHi is a more heterogeneous group that are subdivided ...” should be “... while the NTHi are a”

3, Line 430. Overtuned should be turned over. Overtuned does not mean the same thing as turned over.

Response to reviewers

We would like to express our sincere gratitude for the thorough review of our manuscript titled "In vivo gene expression profile of *Haemophilus influenzae* during human pneumonia." Your insightful comments and suggestions have been invaluable in shaping the final version of our study. We have carefully considered each of your comments and have made significant revisions accordingly.

Our responses are written with red font color, while all reviewer comments are in black.

Reviewer #1 (Comments for the Author):

Polland et al. report the in vivo transcriptome of *Haemophilus influenzae* during pneumonia in human patients. Elucidating gene transcription in vivo in specific clinical niches is of high interest and offers opportunities to identify important pathogenic processes and potentially new interventions for disease states. As such the overall objective of this manuscript is beneficial to the research community. However, there are several issues with the manuscript in its current state.

Specific Comments:

1, The choice of a group level comparison is problematic, and the authors acknowledge in their discussion that pairwise comparison may offer "further interesting discoveries". This problem is compounded by the choice of strains - it is not clear why the specific clinical strains were chosen. Eighteen patients had *H. influenzae* cultured from their lungs. Of these 18, ten samples were excluded due to low levels or poor-quality mRNA or contamination with other *Haemophili*.

We appreciate your suggestion to provide more details on the clinical samples and bacterial strains used in our study. We have updated Figure 1 and Table 2, as well as included additional details in the methods and results sections.

We chose to extract RNA from all 18 respiratory samples in the study in which *H. influenzae* was identified by routine culture at the clinical microbiology laboratory. Unfortunately, mRNA from *H. influenzae* could not be detected in some of these samples, extracted in sufficient amount and quality for sequencing, no library for sequencing could be detected or very few *H. influenzae* reads were detected. RNA is easily degraded and the human samples contain RNases that quickly degrade the molecule into short fragments that are unsuitable for further analysis. Theoretically, this may have introduced a bias in the study and we've added this possible limitation to the discussion.

Additionally, 12 isolates were unavailable for in vitro analysis, although this is unexplained and seems odd given that the bacteria were identified by culture.

All samples were sent to the Clinical microbiology laboratory for bacterial cultures. A protocol was set up with the lab and identified *H. influenzae* were to be isolated and stored.

Sadly, this was frequently missed by the staff at the busy laboratory, and some isolates were thus not saved. However, the consequences of this for the current study are not extensive, since we have limited the main differential expression analysis to the core genome that is shared between different strains.

Figure 1 then shows nine study subjects being analyzed - five both in vivo and in vitro, three only in vivo, and one only in vitro. So only five samples had directly paired in vivo and in vitro derived mRNA, it would have been better to focus on the directly comparable samples.

We agree that a pair-wise comparison would have been desirable, especially as this would have enabled us to go deeper into the genes that were not part of the core genome. With focus on the core genome and “batch” transcriptomic analysis, the difference between using paired or non-paired samples is likely not significant. The samples cluster after environmental factors rather than genetic factors, for instance in Fig 2. Hence, a very similar result to ours would probably have been achieved with completely unrelated HI isolates in the two groups. For follow up studies that are more focused on a limited gene set or one aspect of the bacterial physiology, pair wise analysis can still be done as all raw data are available.

Additionally, the type strain included in the in vitro analysis seems an odd choice since it was originally isolated from a 10-year old patient with acute otitis media. Why not choose a strain isolated from an adult with lung disease like the patient cohort? It is not clear what benefit inclusion of any type strain in the in vitro analysis provides.

Thank you for this suggestion. The inclusion of a type strain with a known annotated genome enabled us to do internal qc and benchmarking of the RNA-seq experiment. At that time, the DNA sequences of the clinical strains were not known (which it is in the revised version of the work). Several studies have used 3655 for mechanistic and genetic analysis. You are correct that it is a NTHi from the upper airway. Compared to many other lab reference strains, it has not been passed in the lab that many times and is still close to strains that are circulating in society. Hence, we were interested in comparing the gene expression of this strain to the clinical *in vitro* isolates (it did not). We decided to keep it in the analysis for two reasons, the increase in N= and for other researchers to be able to relate their results to our results. When this study is published, all raw data sets from the RNA-seq will be made available for others to use.

2, In vitro growth. It is stated that bacteria were grown in complete media and sampled at 240 minutes or in the exponential phase. Where in the exponential phase? Were all isolates growing at the same rate? It would be better to give a specific OD at the point of harvest.

Thank you for this suggestion. We have now added the OD at sampling point to the methods section. The time point to stop (240 min) was determined by a previous experiment, where *H. influenzae* (NTHi3655) was cultured in broth with different inoculation sizes. At 240 min isolates from all different inoculation sizes were in exponential phase.

Also, the bacteria were pelleted by centrifugation and then RNA Later added. Centrifugation can cause transcriptomic changes. It might be better to add culture directly to RNA Later rather than centrifuge the samples.

Thank you for this observation. We have also been considering the effect of centrifugation on transcription, thus making this step in the protocol as short/swift as possible (2 min). An interesting future study would of course be to compare the gene expression of cells pelleted before vs. after the addition of RNA later. A comment about this has now been added to the discussion.

3, The volcano plots (Figure 3 and Figure S3) raise a question. Both have a spot labeled HI_1600, however they are in different locations. Should they not be located in the same position? Or are they derived from different data sets. Figure 3 has a spot labeled *tbp1* and S3 a spot labeled *tbpA*. First the correct gene name is *tbpA* (the designation *Tbp1* is correctly used only for the encoded protein), nevertheless the *tbp1* spot in figure 3 is not in the same location as the *tbpA* spot in S3.

The location of specific genes in the two volcano plots differ because they are based on different data sets (counts based on the core and pan genome, respectively). When comparing the isolates based on the core vs. the pan genome, the gene expression of the two groups receive slightly different logFC's and P-values. This difference occur because the counts are normalized by DESeq2 based on the global gene expression (it uses an empirical Bayes shrinkage for the logFC estimates). See reference below for background:
Love MI, Huber W, Anders S. Moderated estimation of fold change and dispersion for RNA-seq data with DESeq2. Genome Biol. 2014;15(12):550. doi:10.1186/s13059-014-0550-8.

As for *tbp1* vs *tbpA*, see answer below, thanks for pointing this out!

4, The discussion is overlong and too speculative.

Thank you for this comment. We are aware that the discussion is long. To be able to focus the discussion in the revised manuscript we have removed the section about aerobic vs anaerobic metabolism/environment in the lung vs *in vitro*. Consequently, figure 6 has also been removed.

Several other reviewers suggested that we integrated parts of the supplement in the main text discussion. Hence, the discussion is still quite extensive. The nature of the study is hypothesis generating and we have aimed to elaborate about our results and suggest interpretations to add biological meaning to the data.

We have aimed to make it clear in the manuscript when we speculate/hypothesize and when we discuss the actual findings from our study.

Minor points:

1, Line 37. *tbp1* should be *tbpA*, and additionally *fbpA* is not a transferrin acquisition gene rather it is an iron acquisition gene. The incorrect use of *tbp1* for the gene occurs often in the manuscript.

Thank you for pointing this out. Regarding fbpA, we hope that we explained its function adequately in the discussion, but the abstract has now been corrected.

2, Lines 69-70, " ' while NTHi is a more heterogeneous group that are subdivided ..." should be "... while the NTHi are a"

This has been corrected.

3, Line 430. Overturned should be turned over. Overturned does not mean the same thing as turned over.

This has been corrected.

We would like to express our gratitude for your time and expertise in reviewing our manuscript.

Reviewer #2 (Comments for the Author):

Review of "The Haemophilus influenzae in vivo gene expression reveals major clues about bacteria central metabolism, acquisition of trace elements, and other essential pathways during infection of the human lung"

By Linnea Polland, Yi Su, and Magnus Paulsson

Summary: The authors carried out the first global transcriptome analysis of Haemophilus influenzae (Hflu) bacteria directly from human lung infections. Sputum or bronchial wash specimens from pneumonia patients were screened by standard clinical micro culture for Hflu, and where found, an isolate was collected for in vitro RNA-seq, and total RNA from the banked clinical specimen was processed for in vivo RNA-seq. In the end, the authors produced core gene transcriptome profiles of Hflu cells living within 9 cases of pneumonia infection, and they compared these to in vitro transcriptome profiles from 9 matched clinical isolates in exponential phase in rich medium. The in vitro grown cells all had relatively comparable core gene transcript profiles, whereas in vivo transcript profiles were distinct from the in vitro samples and had higher within-group variance. A DESeq2 analysis identified genes that were differentially expression in in vivo samples compared to in vivo. Specifically, differentially expressed transcripts were identified with: (a) reduced relative expression encoded proteins involved in energy production, and (b) increased relative expression encoded proteins involved in stress responses and nutrient acquisition.

General assessment:

This is the first analysis I've seen to measure global gene expression profiles of Haemophilus influenzae directly from human infections, and I believe this is an important dataset to release. The authors have put in a large effort and expended many resources to acquire this dataset, and their analysis provides insights into bacterial gene expression within a human infection in contrast to typical in vitro growth conditions.

However, I have three large issues with the manuscript as is: (1) the MS reads a bit like a draft and needs substantial editing for grammar, style, clarity, and content; (2) the genomic and bioinformatic methods and QC steps are poorly described, so some of the major elements of their approach are obfuscated; and (3) the analysis emphasizes how the expression profiles in human infections differ from 'standard in vitro conditions' but they don't contextualize this 'standard' condition as log-phase shaking culture in rich media in their discussion. This context makes some of their findings a little more straightforward, especially if compared to some other Hflu transcriptional regulation, RNA-seq/microarray, and Tn-Seq experiments that have been published. Addressing these issues would improve the MS considerably without necessarily carrying out new or different analyses.

Main comments:

This MS needs a strong English language edit.

Thank you for this suggestion. We have had the manuscript edited by a language editing service and we hope that it is now improved in this regard.

In addition, the motivation and purpose of many parts of the analysis are not provided, and the description of the analyses themselves can be vague, even with the supplementary files. Including stronger reasoning for the analysis approach would help make this work more digestible. It is possible that the authors need to get more detailed information or consult with whomever helped them with the initial data processing steps to go from reads to count tables (UPPMAX?)

More description and results associated with the in vivo experiment are needed, especially related to the fact that most sequenced RNA molecules were not from Hflu. This is not a surprise but since data like this have not been reported before, it is important to discuss the results that lead to the Hflu transcript profiles. The Supplement Table S3 needs to include total reads mapped to their "pan-transcriptome" reference for Hflu. There appears to be a very wide range of read counts per transcriptome, so this needs to be reported.

The extremely large difference in useable read counts between in vivo and in vitro is also notable, and it may affect aspects of the analysis.

Thank you for these useful comments and suggestions. We have moved the bioinformatics methods section from the supplement into the main text methods and edited it with the help of a bioinformatician and language editing service. We have also included a discussion about the limitations of our method in the discussion.

We have now also created new tables included in the supplement with full read counts for each sample and gene. From these tables, the read counts should be clear for the reader. We have also included this difference as a possible source of bias in the discussion. In addition, we've included both bas code and r scripts in the supplemental data.

It would also be useful to report on how well the rRNA depletions used worked.

Great suggestion. rRNA counts have been summarized in the results section and added to the revised Supplemental table S3 and summarized in the result section. rRNA depletion was more efficient on the clinical samples than on the bacterial strains.

Descriptions of the bioinformatics approach were convoluted by dividing up information between main and supplement, but then these together also did not adequately explain how the raw data were processed into count tables. I could infer what was done, but this raised complex issues. I don't think all the analysis issues necessarily need to be fixed but the authors should acknowledge the strengths and limitations of their bioinformatics approach:

-- I understand why a pan-genome reference was used, and how Roary output was used to build a "pan-transcript" reference for Kallisto, how Kallisto was used to count, and how the

initial count tables were imported for DESeq2 analysis, etc. However, none of this is described.

Thank you for pointing out the need for us to give more details about this process. This interpretation is correct and we hope that these processes are clear and easy to follow in the revised methods section, together with the R scripts in the supplemental data.

-- The authors need to acknowledge and consider the types of read counting artifacts that can arise from their approach. This is especially true because of how they do their Kallisto-based counting.

The pseudo-alignment done by Kallisto introduces some noise in the read count because of misclassification between genes and even between species with similar genes. However, Kallisto is very resource efficient which is a great asset when working with large sequence files. We don't think that the consequence for prokaryotic transcriptomic analysis by this approach is as profound as for eukaryotic analysis, as for instance splice variants in eukaryotic cells can't be evaluated using this approach.

-- What is the purpose of the control 3655 strain? It's not even in the pan-transcript reference to check how this approach might be affected by sequence divergence among strains.

The inclusion of a type strain with a known annotated genome enabled us to do early internal qc and benchmarking of the *in vitro* RNA-seq experiment. At that time, the DNA sequences of the clinical strains were not known and we had no core or pan genome to work with. Several studies have used 3655 for mechanistic and genetic analysis and it is a NTHi from the upper airway (acute otitis media) of a child. Compared to many other lab reference strains, it has not been passed in the lab that many times and is still genetically and phenotypically close to strains that are circulating in society. Hence, we were interested in seeing if the gene expression of this strain differed widely from the clinical *in vitro* isolates included (it did not). We decided to keep it in the analysis for two reasons, the increase in N= and for other researchers to be able to relate their results to our results. When this study is published, all raw data sets, including read counts, will be made available for others to use and compare their data and in that setting we believe that 3655 may be useful.

-- Are there genome assemblies of the clinical isolates? This appears to be stated in the supplement, but I don't think this is what the authors mean? It would obviously be much better if their pan-transcriptome was made from annotated assemblies of the actual strains they used in their experiment. They know the MLSTs and there are hundreds of Hflv genomes in public databases, so they might've been able to match up their reads to better references. Why were the 15 used chosen? Why wasn't 3655 included? If they'd used many more, they could have used an even more reduced core gene list.

Thank you for this question about assemblies and the core/pan genomes. The reported MLST were based on transcriptome assemblies. There are limitations in determining MLST on the transcriptome so we have since sequenced genomic DNA from the clinical isolates

and added the updated MLST to the revised Table 2. We reasoned that the 15 strains in the KEGG database would be a good balance and give a reduced core genome, while the pan genome would still be limited. As we didn't have the genome of the clinical strains this was not an option at the time. Using an external set of input genomes has the strength that we can use the same core genome in follow up studies. However, we look forward to do pairwise comparisons using the entire genome of each strain in those studies. We have added a discussion about this in the revised manuscript.

-- Even without redoing any analyses, it would be worth discussing or commenting on the strengths and limitations (especially with respect to accurately counting reads) of their approach.

I believe that this question is about the pseudo-alignment and a section about pseudo-alignment limitations has been added to the discussion section.

-- Which core genes were used to do the GO analysis? The GO analysis sections also need heavy editing for clarity and meaning.

Thank you for pointing this out. In the revised manuscript we now clarified how many and which genes were used for the GO analysis of biological function.

-- How were analyses done when including all genes? Were missing genes treated as zeros or NAs?

Regarding the DESeq analysis: genes with no counts in a sample were counted as 0. Genes with a total count (all samples taken together) of <10 were filtered from the analysis. If further analyses using the pan genome were to be done, this approach would have to have been modified.

Detailed comments:

Lines 2-4: Cumbersome grammar and very long for a title. Maybe something like: "In vivo gene expression profiles of Haemophilus influenzae during human lung infection reveal..."

Thank you for this suggestion and we agree that it was a bit long! The title has now been changed.

21-43: Expand to explain more about slightly more about data acquisition and analysis pipeline, since this is non-standard due to diverse strains. E.g. what fraction of sequence reads were bacterial in the in vivo experiment?

We have improved the methods section, added data about each sequence file, including rRNA and accession numbers in the supplementary data. We have also produced a new table with Kraken2/Bracken data with the number of reads that were left after human depletion, assigned to HI and the top5 bacterial species in the samples. This has been summarized in

the results section and full tables are available in the supplement.

47: *in vitro* and animal studies. Worth emphasizing here and elsewhere that animal models of *H. influenzae* lung infection are relatively poor.

This has been emphasised in the revised version and a short notion of this is also found in the background.

47: "Avoid"? Or to understand the biases? How would these results compare to "dual RNAseq" of NTHi+host cells?

This paragraph has been revised. A comparison to the dual RNA-seq data set is of high interest, but outside the scope of the current study.

50-51: How is this quantifiable from the PCA?

This is a good point. The variation can be quantified, but for this purpose we do not think it is necessary. We have revised the sentence in the importance section. However, we think it is safe to say that the unsupervised clustering in the heatmap of FIG 2 (of the core genome) shows that two groups consisting of *in vivo* vs *in vitro* bacteria has formed.

52-55: How does this relate to how "vital" the pathways are?

It does not and this has been rephrased.

62: "evolutionally developed" to "evolved as a"

This has been corrected.

63-71: Genomic details are of some interest, but this is awkward. How does this fit with the aims here? Given that most of the genomic heterogeneity is ignored, this introduction might better focus more on the inadequacy of *in vitro* and animal models, though it's worth mentioning the high diversity. Unclear if a discussion of capsule is really warranted much here.

Thank you for pointing this out. We have put less attention to serotypes in the revised manuscript and focused the introduction more on the models as suggested.

72-83: The Pa example is fine; probably worth citing some others that also show this sort of thing. This could also use a discussion of animal models of NTHi lung infections, for which there are some. Also some literature on bacterial-host cell that might be of interest, or at least further expand in the discussion.

Since our draft was published a new study on *H. influenzae* transcriptome in mice has been published. A short comment about that and about some of the dual-seq studies has been added in the introduction.

97-113: Should this refer to Supplementary Table S2?

Yes, the patient population is presented in supp table 2 and the reference is found in the first part of the results section.

119: taxonomic analysis of genome assemblies using Kraken2?

That was incorrectly described in the first version. Initially taxonomic analysis was done on transcripts, but now that we have gDNA assemblies instead this has been reanalyzed. A new section about this has been added to the methods section.

121: Why was this reference chosen?

This was mistakenly put here and has now been removed.

125: Were all strains in exponential phase after 4 hours? Approximate ODs? Growth curves may be useful here. Choice to use duplicates should be specified.

We have now added more details about the culturing conditions to the methods section, including OD values. The time point to stop (240 min) was deduced from a previous experiment, where *H. influenzae* (NTHi3655) was cultured in broth with different inoculation sizes, and the log OD change was calculated from OD-measurements at several time points. At 240 min isolates from all different inoculation sizes were in mid- to late exponential phase.

133: These supplementary methods might as well be added to the main MS.

This has been done.

133: Supplementary Table S3 is incomplete. What is the total number of Hflu reads mapped for each sample? What are the public accession numbers of

The tables in supplement have been revised with accession numbers and a new table with taxonomic data has been added. In addition, count tables have been made available as well.

136-144: Is this section out of place? Supplement Tables S2 and S3 belong before. Table S3 is incomplete.

The methods section has been restructured in the revised version. Supplement table 2 and 3 contain results so we refer to those in the results section.

148-150: Might as well move the missing steps out of supplement and into the main, or at least give a short version. It is key to specify

These methods have been moved into the manuscript.

Supplement File S1 Bioinformatic methods: This is a bit of a mess. May the authors need help from UPPMAX? It implies there are also genome assemblies of the individual isolates? If so, these methods and datasets also need to include explanations of the genome assembly process. If a genome assembly was performed, why weren't the seven strain isolates included in the pangenome use with Kraken for the gene counting? Or was there no genome sequences, and these are just using the pan-genome calculations with the reference strains to count these RNA seq reads with Kallisto? Or were there transcript assemblies? There are a lot of missing moving parts here that strongly affect interpretations, etc.

We have put much effort into improving this section together with bioinformaticians. We hope that you find it to be improved. Initially we did not have access to the genome assemblies, only transcript assemblies.

150-153: This needs to explain what was done with non-core genes.

We think that the reviewer refers to genes with low or zero counts in the pan genome. Only genes with in total 10 or more counts were included in the analysis and samples that had gene counts of 0, were included as 0 in the analysis. If we aimed to continue the pan genome analysis, this approach would have to be evaluated as 0 can mean that a gene was present but not expressed, or that it was not present in that strain. For the core genome analysis, we believe that this strategy is robust.

163: All core genes, or representative core genes from a specific reference genome?

All genes in the core genome, this has been clarified in the main text.

171-172: Fine if no multiple testing correction but should specify p-cutoff.

This was missing and has now been added.

175-177: Supplement Table S3: This is inadequate. Please include total H. influenzae reads mapped. Please include individual accessions to samples. Please include final count table. Please explain why some files have R1 and others R1 and R2? How was the difference handled in the analysis? It would be desirable to include the count table as a supplement if possible, or deposit and EMBL or NCBI as a gene expression dataset.

New tables has been created as per your helpful request, including full count tables. We have also explained the paired-end and single-end FASTQ files in the methods.

184-187: This might refer to Supplementary Table S3. What are the QC criteria? What is the mean and range of read counts from the in vivo samples? I believe that there's likely enough Hflu reads here for these comparisons, but the read counts must be relatively low, especially compared to the in vitro samples. This might affect some aspects of downstream analysis (e.g. comparing transcript profiles from 200K reads to 20M reads). Importantly, % of Hflu reads that are from rRNA would be also helpful to know. This gave specifics for all read files, but it would also be useful to have these stats aggregated by sample.

QC criteria included both quality of the RNA (RIN >2), a detectable sequencing library and then the sequencing. If less than 1% of HI reads in pilot sequencing, we did not go ahead with the sample for further sequencing. Both full read count tables a summary in the results section has been added to the revised manuscript.

188-191: The exact condition needs to be specified, that these are cells growing in rich media and are in exponential phase. The dramatic differences between these and the *in vivo* profiles are quite interesting, but other *in vitro* conditions may look more similar, for example in a nutrient deprived culture.

We have now specified the culture conditions in the methods section. Bacteria cultured *in vitro* using e.g. a defined or minimal media would possibly have more of a transcriptomic resemblance to *in vivo* isolates. One of the things we aimed for was to show that bacteria cultured under typical conditions in which *H. influenzae* is studied, e.g. in BHI broth, differs in behavior from bacteria growing in the lung. This may affect many aspects of the bacterial phenotype, including susceptibility testing. An interesting follow-up study would be to compare the transcriptome of bacteria growing in different media to the clinical samples. A brief discussion about this has been added to the revised manuscript.

200-206: Explain the logic. I think I understand, but the implicit use of this reference with Kallisto for the RNAseq experiment suggests reasonable logic. There's a lack of clarity about whether the cultured isolates also had genome sequences and why these weren't included. If these are all NTHi, was it important to include typeable strains? Why only these 15 strains? Wouldn't knowing the MLST at least help pick appropriate references? Was Roary used with all default settings? Why isn't NTHi3655 included in the study?

This section in the results section has been expanded and a discussion about the strain selection has been introduced. As the core and pan genomes were created, it was interesting to note that the core genome was very moderately reduced for each added strain, while the pan genome continued to expand.

206-212: This is too packed up and needs to be detailed. How are reads counted against the pan-genome? Are only core genes included in the analysis? Could there be any issues mapping reads from some of the isolates to the pan-genome that might affect the analysis? It would be worth acknowledging these issues.

In the revised manuscript, we have summarized the data also for the pan genome. However, because of the large differences in the supra genome between the strains in the study, as well as the strains used to create the pan genome, we do not think that this analysis is meaningful. One reason for this is the use of zero as zero and not NA, as has been discussed elsewhere.

211-214: This is all very interesting but is stated without interpretation. Worth stating that this contrast is against exponentially growing cells in nutrient rich conditions. An outcome would also be worth stating.

It probably makes more sense to present the PCA first, since the DEG analysis depends on this to a large extent.

Thank you for these suggestions. We have now moved the PCA and heatmaps to ahead of the DEG analysis results and added information about the laboratory conditions in this paragraph.

Also as a note of caution, one assumption of DESeq2's default normalization is that most genes aren't differentially expressed. Did the DEG analysis consider using clinical source (subject) as a co-variate in an LRT to help account for the heterogeneity among the bacterial genomes?

"Subject" was included as a co-variate. We have added the R script to the supplement so that all analyses are properly described and can be replicated. A brief notion that this was done has also been added to the results section.

217: The PCA might belong prior to DEGs, since it clarifies how the samples do/do not cluster by genotype and condition.

Yes, we have restructured and reordered these sections.

224: Is the heatmaps only of high-variance, or does it use the DEG analysis? Clarify.

The heatmaps were of the top most variable genes of the core genome, based on normalized data: the DEG genes are therefore not mentioned.

225-227: This grammar implies that the clinical isolates are distinct from the lab-grown ones, when in fact, the authors are comparing Hflu transcript counts from lung samples versus from in vitro log-phase samples using a strain isolated from the same lung samples. This is very different. What's meant is that the in vivo samples showed higher heterogeneity than diverse strains grown in a single condition. This shows that a lot of the heterogeneity is probably driven by how variable the conditions are, rather than the genomic heterogeneity among the isolates grown in broth.

This is correct. We have revised the results and discussion and we think that this is clear in the new version.

227-228: How are accessory genes treated in the pan-genome heatmap? As zeros? Why would the clustering differ, and what might drive these differences?

The genes not expressed (or present) in the core and pan genomes respectively were treated as zeros in the analysis, which has now been specified in the method section. The clustering will differ depending on the use of 0 as zero or NA, and the effect of this on clustering when the genetic heterogeneity is large may be dramatic. If further pangenome analyses were to be done, this would have to be taken into account. However, the effect of this on the core genome is not very extensive since the core genome is expected to be shared between the strains.

229-233: This is a confusing presentation. How was the analysis conducted? Were gene absences treated as NAs? As zeros? If the latter, this would cause a lot of oddities. The focus on core is reasonable, but also there are so many artifacts that could be arising during the Kallisto-based mapping.

The absences were treated as zeros, both in the pan and core analysis. The reviewer raises an interesting and highly relevant topic: how to handle missing genes in RNA seq data. Missing genes of specific samples/isolates could mean several different things: the gene is truly not present in the genome of the isolate, the gene is not expressed in the isolate, sequencing depth/method etc. Using NA's also introduces a bias in your analysis, giving less weight to missing genes that might only not be expressed in a certain sample/condition. One must thus choose between two not optimal methods.

Since this is indeed interesting and worth some reflection, we have added a notion of the number of zeros in the core genome in the MS. One of our top DEGs, CGSHiII_03600, was one of the genes having the most zeros (six out of 22 sequenced isolates, all *in vivo*): interestingly, and maybe to illustrate our point, four out of these *in vivo* isolates had *in vitro* isolates showing expression of the gene, making it more likely representing true absence of expression rather than of the gene itself (or of course, a sequencing or RNA-extraction artefact).

235: Organizationally, still makes sense to bring DEGs down here, since this next section is about those genes.

This has been done.

237: Stability within-group was not shown. Perhaps plotting individual genes between conditions would clarify. I suspect the variance within the *in vivo* samples will generally be much higher than within the *in vitro* samples, even if there are still statistically significant differences.

Thank you for this suggestion. A plot has been added showing the counts of the top 30 most differentially expressed individual genes (Figure 4). A qualifier ("more stability") has also been added to this sentence.

242: Higher relative expression.

This has been corrected.

244: What is meant by "interaction"? Correlated expression with *msrAB*, or some other unspecified link? What is meant by "connected" in this context?

These genes were "predicted functional partners" in the STRING protein interaction database (a reference that has now been added), which in turn uses extracted data from databases Biocarta, Biocyc, GO, KEGG and Reactome. This has now been specified in the main text.

246-247: Strong evidence support a role for these genes in transformation but also their regulation is known to depend on depletion of preferred sugars and purine precursors.

Yes, thank you for this observation! We got interested in the competence mechanisms of *Haemophilus* during the analysis of our data and have learned that there is a connection between competence and nutrition status in this species. This is elaborated in the discussion and another reference to the function of these specific genes (*dprA* and *rec2*) was added.

251-252: For all DEGs or from all genes? I think this must have used some specific reference genome and reduced to only the core protein-coding genes? What fraction of genes got GO annotations?

This information has been corrected. All core genes were entered into Panther. 1061 of these were annotated, and 896 had a biological process assigned.

255: Please comment on genes of unknown function or genes not annotated by Panther.

Only 6 genes could not be annotated by Panther. 165 genes could not be assigned a biological process. The GO annotation is limited by the database that is used, in this case PANTHER. This is a potential limitation for the biological interpretation of the data and we have added a comment about this in the discussion.

260-263: Grammar is convoluted. These processes were not "less common".

This has been rephrased.

263-264: This is a good follow-up. What is the obvious interpretation?

Thanks. We believe that the obvious interpretation is that the bacterial cells in the lung have a lower metabolic activity, and this has been added to the text.

264-267: Compared to shaking culture that should be well oxygenated?

Yes, but due to space limit, we were forced to cut the discussion and some of the results about anaerobic vs aerobic environment in the lung. This will be addressed in a follow up study that is specifically designed for the purpose of studying anaerobic conditions.

272: "Less common" is strange grammar. The fraction 2 of 11 means something different than implied here.

We have now removed the fractions since it was confusing for the reader. Less common has now been changed to "enriched in the downregulated DEGs".

277-285: Grammar and interpretation are still complicated here. Given that several of the

genes discussed above are in these processes, connecting the "most different" DEGs and these pathway analyses would probably help.

See answers above, grammar has been corrected.

286: No measurements of Mo-transport were performed. Clarify.

Correct, this has been rephrased (line 446).

298-299: Excellent goal. Worth specifying that the condition chosen was rich-media shaking (well oxygenated) during exponential phase.

We have specified the laboratory conditions we used in several parts of the manuscript.

299-301: Thank you. This is great!

Thank you for this kind comment.

304-305: This is primarily due to the default paralog splitting done by Roary. Homologs merged would give a smaller cloud. This is minimized by use of complete genomes. Why were those the choices, and what is the relevance of that pan-genome with respect to the goals of the paper? The use of this for mapping seems like it was the main purpose, and for defining a core genome. But those aren't really discussed. Using a Kallisto-based approach was an interesting, potentially cool decision but has some implications and potential counting artifacts that aren't discussed. For example, even the reference genome wasn't included in the pan-genome reference, so the ability to even detect counting artifacts wasn't really feasible.

Thanks for pointing this out. The purpose of the pan genome was to extract a core genome, and the purpose of the core genome was to map reads with Kallisto. This has been clarified in the methods, result and discussion. The use of Kallisto is discussed elsewhere, but it is clear that it introduces some noise in the analysis.

305-308: This assumption is probably true, but no evidence is presented for this.

True, we've added an example from the clinical strains. Only two isolates, LUIN_31 and LUIN_33, expressed the bla gene coding for the beta-lactamase TEM-1.

310-312: Well, that study was using a single strain, so included all its genes. Although the proportion is distinct, the power was also probably quite different in those lab experiments.

We made a note about this in this section (line 527-533).

315-318: These are fantastic observations, but I might re-word or re-think this. I would argue that only the in vitro grown cells show strong clustering. Sure, the in vivos are more to one side of the in vivo cluster in the PCA, so there's nice strong differential expression, but there's still clearly a great deal of heterogeneity here in the in vivo samples. This could be

because of many reasons: e.g. much lower sampling of reads in the Hflu transcriptome, no opportunity for replication, not sure the infections were monoclonal, unknown genotypeXenvironment interactions that are irrelevant in log-phase broth culture, etc. These would be interesting to discuss.

All these points are very valid. This section has now been rephrased (line 535-548), a notion of the reasons of the more heterogenic appearance of the gene expression of *in vivo* bacteria has been added, and we think that this is now more correct.

331-338: These are great speculations, and potentially biofilms are important, but also other types of nutritional shifts, like simply stationary phase, lower oxygen, etc could be important. In particular, I would guess that CRP response is on, and relative induction of purine biosynthesis and competence genes suggests also suggest a nutrient limited state.

Thank you. We adopted some of these examples into the text.

342-343: Wasn't the RNA from liquid cultures? Were the shaking cultures also in high CO₂ or is this just how the plates are grown?

350-352: It's a smaller fraction used for recombination but regardless, the state of competence is induced by specific nutritional conditions. Might be worth noting that the competence genes have been shown to be involved in biofilm formation.

They were from liquid cultures with 5% CO₂. However, due to space limits, the discussion about anaerobic growth has been removed.

363: Another competence connection.

Yes! Another reference to competence and purine depletion in the above section (Sinha et al 2013).

373-375: Worth citing/discussing *in vivo* mouse lung Tn-Seq experiments + and - flu infection, as these also hit purine biosynthesis and some others seen here, so provides a functional connection.

We have added a Tn-Seq experiment of *P. aeruginosa* and *E. faecium* showing importance of purine/pyrimidine synthesis genes to the references in this section (one of H.i. was not found).

376-391: Good discussion but might also bring up relevant old microarray experiments that find this and related iron acquisition genes differentially regulated in various *in vivo* conditions. I am also a little lost about directionality. Which setting appears to have more access to oxygen, based on the gene expression analysis?

We have now removed the discussion of anaerobic vs aerobic. We have added a reference to an *in vivo* microarray study of H.i. with regard to these genes.

376-403: It is important to remember that all of this is in contrast to growth in rich media under non-nutrient limiting conditions. It doesn't seem surprising that either iron or Mo are going to be way more limited in lungs than in rich media at log-phase.

We agree that many of our findings are intuitive. We believe that we have described the growth condition studied *in vitro* sufficiently in the revised edition, and that our aim was to compare H.i. *in vivo* to isolates growing in routine growth conditions that are commonly used for culture of H.i.

404-410: This seems unneeded? Both of the next paragraphs are about redox, which related very tightly with previous discussion of TCA, iron, and oxygen.

Good suggestion, it has now been removed.

411-428: Cool discussion. Neat findings.

Thank you!

429-441: This could use expansion about the issues of doing this, besides RNA quality. That's fine. There's no poly-A selection here so that should be a major problem. But the massive amount of human RNA is definitely a problem for getting this type of work done, and the few useable reads that emerge present some difficulties of interpretation. What about the bioinformatics problems associated with use of the reference pan-transcriptome?

Thank you for this suggestion. A discussion about rRNA and read count balance has been added. The pan transcriptome is now discussed in an earlier section.

Allelic variation is also very high among Hflu strains, so read mapping (be it actual or quasi-alignment) may undercount for some alleles versus others, especially since the strains used in the experiment weren't the strains used in construction of the reference pan-genome.

This is true and a background noise may have been introduced that affected the analysis. This could be avoided with a completely different strategy (pair-wise comparison of *in vitro* vs *in vivo* of the same isolates). We will consider that when designing follow up experiments. However, pair-wise comparisons have many limitations as well, for instance the lack of biological/technical replicates for the *in vivo* samples.

429-441: What about the possibility of polyclonality in infected samples?

This cannot be excluded and has been added in the discussion.

455+: I think a key outcome is that we have a better idea of the environment experienced by Hflu when in the lung, and it is possible that comparing this data to other data in other *in vitro* settings would look closer to these real samples, so that we can better mimic *in vivo* experiments.

Thank you! And to this reviewer we would like to extend our greatest gratitude for your time, your insightful comments and your very comprehensive, relevant and helpful review. We acknowledge that this review took time to compile but with the help of your comments this manuscript has been much improved.

Reviewer #3 (Comments for the Author):

I have read the manuscript with great interest, and I think this is a good example to show that in vivo tests are not always translatable to the in vivo situation. The study setup is clear and even though there might be a bias because of the large number of samples that did not meet the quality controls, the in vivo transcriptional data is valuable information.

I'm not familiar with the current rules regarding data sharing, but I would encourage the authors to also share their R codes to enable replication of the data and control on the data analysis pipeline.

Thank you for this suggestion. We have added all bash and R code to the supplemental data.

In this discussion line 440 is mentioned that samples with less than 1% transcript assigned to H. influenzae were excluded. Could the authors include a supplemental table of the percentage H. influenzae transcripts in comparison to total transcripts, and also a list of other pathogens with percentages found within the transcripts? This could give some inside which co-infections were possible present.

This is a great suggestion. We've made a new supplemental file 4, in which the taxonomic data is shared. In this file you'll find the number of reads that are assigned to "bacteria", to HI and then a list with top 5 other species found in the samples.

I would prefer to include supplemental file 1 into the material and methods section of the main manuscript.

This is a very good suggestion, and this point was suggested by several reviewers. We removed supplemental file 1, improved the text and integrated it into the methods section.

Although the discussion is already extensive (but very useful), I would prefer to include the discussion of supplemental file 5 into the main manuscript.

Due to word limits, we had to restructure the discussion. We integrated supp file 5, but unfortunately we had to cut some parts, for instance about anaerobic respiration. We will have to come back to this in future projects.

There are track changes present in supplemental file 1.

Thank you for taking note of this!

Supplemental file 5 contains some minor textual error, such as a missing space between H.influenzae in the first and third sentence.

Thank you for pointing this out. Some parts of supplemental file 5 have been integrated into the main text and the rest have been edited in the revised version.

Reviewer #4 (Comments for the Author):

This manuscript by Polland and collaborators compares the transcriptome of Haemophilus influenzae in vivo with that in vitro. Although a large number of patients were initially included, only a few samples were ultimately analyzed. Therefore, the work does not provide sufficient evidence to draw a general conclusion regarding the metabolic status of NTHi in vivo. This is due to the limited number of samples (n=9) tested in vivo, which had varying patient statuses, as well as the small number of reference genomes (n=15) used to create the limited core genome that formed the basis of the DEG analysis. However, the PCA plots in Figure 2 present an alternative view. Unfortunately, the data novelty related to in vivo infection is quite limited when compared to previously published models (such as those involving animal/primary cells and cell lines).

Thank you for your comments. We believe that the novelty is quite high as this is the first description of H. influenzae gene expression in humans that we are aware of, and H. influenzae is a strictly human pathogen. Therefore, animal models are of limited value. However, since this manuscript was submitted, a very interesting study of the in vivo gene expression of HI in mice has been published (and we refer to this in the introduction). There are also interesting transcriptomic reports from cell line and other models that we discuss and compare our data to.

Regarding the number of samples, this is the first time anyone has done (or at least published) this kind of analysis for H. influenzae. One reason for that is that the samples have to be collected specifically for RNA studies, another is that it is difficult to extract RNA of adequate quality and a third is that each sample has to be sequenced with several 100M reads to get enough reads for the analysis. We agree that it would have been great with more samples and as this technology is becoming even more available and affordable we are convinced that more and larger studies will be performed.

1. Line 21. Haemophilus influenzae is not commonly associated with HAP. This should be corrected in the Introduction.

Thank you for pointing this out! This has been corrected (changed to “and COPD-exacerbation”).

2. The authors should acknowledge previous studies on H. influenzae transcriptomics in animal models.

Thank you for this comment. Through-out the manuscript previous animal studies on H influenzae transcriptomics is referenced and discussed. Of particular interest is the new mouse study that was recently published by Junkals group (<https://pubmed.ncbi.nlm.nih.gov/37195232/>).

3. Line 79. Pseudomonas aeruginosa does not belong to the same species and has a considerably larger genome than H. influenzae

Thank you for this comment. This is correct. Similar studies have been performed on *P. aeruginosa* and therefore we believe that it is of interest for the reader to be introduced to the transcriptomic analyses on *P. aeruginosa*.

4. Line 84: It is unclear what "complicated" refers to, and the authors should provide more context or clarification.

Thank you for pointing this out. This section has been altered and the word complicated removed.

5. Line 86: The *H. influenzae* *in vivo* transcriptome is not unknown (Aziz is cited in the Discussion and should also be cited in the Introduction).

<https://www.frontiersin.org/articles/10.3389/fcimb.2021.723481/full>

<https://doi.org/10.1016/j.csbj.2021.05.026>

<https://www.frontiersin.org/articles/10.3389/fmicb.2019.01622/full>

Thank you for your comment. We have described more clearly in the revised manuscript how our study was performed and this sentence has been modified. What we refer to as *in vivo* condition is the gene expression as it is while the bacterial cells are inside the human lung. It is true that Aziz et al used isolates from samples collected from nasopharynx and BAL-samples, but RNA in these samples was not preserved and analyzed. Instead, *H. influenzae* isolates were cultured in specified laboratory conditions, and RNA isolated and extracted from these *in vitro* condition cultures.

Ackland et al used an macrophage model based in the laboratory, and so *in vivo* transcriptome studies were not performed by these authors. However, this is an interesting study and it is now referred to in the introduction.

The last reference mentioned above was a nice review in which several studies on the *H. influenzae* transcriptome was mentioned: out of 10 reviewed studies of the gene expression of *H. influenzae* (most of which are referenced in different parts of the current manuscript), 8 were *in vitro* studies of various kinds and 2 were *in vivo* animal models (1 murine, and 1 chinchilla model)

In the revised manuscript we have added some of the above mentioned references in the introduction, with an extended discussion of previous *in vitro* and animal studies.

6. Line 88: Several studies have shown a significant discrepancy between *in vivo* and *in vitro* conditions; thus, it is a fact rather than a hypothesis.

We believe that this has not been shown for *H. influenzae* cells that grow in the human body.

7. Line 180, and Table 1 and 2: The sample description is unclear and confusing, particularly when attempting to relate it to Figure 1. Therefore, the figure legend for Figure 1 could be made more descriptive and clear.

Thank you for this helpful suggestion. We have improved text, legends, table 2 has been reworked and fig 1 has been improved. We think that it is easier for the reader to understand which samples that were used and why.

8. Line 187: It is unclear what is meant by "picking colonies" from routine cultures from only 6 patients when samples from 8 patients were analyzed for transcriptomics. It is confusing why an equal number of samples were not chosen for downstream analyses. Additionally, why were only two colonies picked, and did these colonies undergo WGS?

Unfortunately, only 6 of the 18 included samples with growth of *H. influenzae* were saved by the busy routine microbiology lab, despite extensive work with the study protocol at the microbiology lab. We would have preferred to include only paired *in vitro* and *in vivo* isolates. However, since we analyse the core genome, we believe that the difference between analyzing paired samples or unrelated samples is small.

Duplicates of *in vitro* sequencing were included as replicates to limit the effect of pipetting errors. It would be preferable to obtain biological replicates also for the *in vivo* samples, but this would be very hard to obtain.

9. Line 189: Strain "3655" is not described in the manuscript. It is unclear why it was included in the study since it is not sequenced according to Supplementary Table S1.

We included the reference strain 3655 as qc for our initial transcriptomic analysis since 3655 has a publicly available genome to which the RNA reads could be mapped. 3655 is originally a clinical strain collected from the upper airway from a patient with acute otitis media. As compared to many other "lab-strains", it has not been recultured as many times and is still close to circulating NTHi in its phenotype. By including this, other researchers can relate their data to our data set if they wish. All raw data used in our study is available in the supplemental files, at the ENA or PubMLST.

10. Line 192: Although the manuscript states that eight patients were analyzed, Table 1 shows nine patients in the "study cohort," which is confusing.

The reviewer is correct that the inclusion of study subjects and samples are somewhat confusing. To summarize: 9 subjects/patients were included, together providing 8 *in vivo* isolates and 6 *in vitro* isolates. This has now been corrected in the text, and figure 1 and table 2 has also been altered.

11. In general, a cohort is defined as "a group of persons, usually 100 or more in size".

This has been corrected.

12. Line 202: Why were capsulated strains (small in number) included when you were searching for the core genome with a high number of NTHi strains? I think you should only focus on NTHi, excluding the capsulated strains, especially when you did not have a good number of capsulated strains (only 3); the sample pool is rather weak here. The information

on core genomes of NTHi (various clinical isolates and reference genome, more than 12 strains) has been widely published in several studies and should be referred to in your study.

We have improved how we generated the core genome that we used and improved both the methods and results sections that describe how and why we did this. We reasoned that the 15 strains in the KEGG database would be a good balance and give a fairly reduced core genome, while the pan genome would still be limited. As we didn't have the genome of the clinical strains this was not an option at the time. Using an external set of input genomes has the strength that we can use the same core genome in follow up studies. However, we look forward to pair-wise comparisons using the entire genome of each strain in those studies.

13. Line 211-214: It is unclear what the difference is between the 328 genes of the core genome (defined as DEG) vs. the total core genes (1067). "In total" of what? Does this refer to the 9 isolates and 3655 reference genome (line 205)?

Throughout the study, the term DEGs are used to describe differentially expressed genes that are statistically significant (p below 0.05 and fold change ± 1). The "in total" refers to all genes in the "core gene" subset.

We have made some alterations of this text and we hope that it is now more clear as to what we mean by DEGs, core and pan genome.

14. Figure 2 is confusing regarding the clarification of in vitro isolates (line 187). I assume those with "_1 and _2" are 2 colonies picked from 6 patients for in vitro analysis. The authors need to make the indication clearer in the figure legend. Why did the in vitro sample of HI_LUIN_26_1 and _2, even when in in vitro grown conditions, not cluster together and instead showed a very discrepant heat map profile, despite being from the same isolate, compared to other _1 and _2 isolate pairs?

Thank you for pointing this out. We have clarified the use of the _1 and _2 in the figure text (and also in table 2).

We don't know why _26 did not cluster together with its counterpart, but this was also the reason for us to use duplicates. These duplicates are not just technical repetitions of sequencing, they are based on different colonies from the agar plate that were independently cultured. There might have been a pipetting error, it may be an effect of biological variation or (less likely) the sample may have contained multiple clones of the *H. influenzae*. We do not know how extensive differences in the global gene expression that small variations in growth conditions cause (more or less biofilm? variations in the broth composition? Small variations in temperature?).

15. Line 225, "in vivo samples of clinical isolates" instead.

This has been corrected.

16. Line 227. Figures 2C and D are difficult to comprehend and should be simplified by using different color labeling instead of both colors and names for different strains. The same

applies to Figure 4, where the use of different colors for strains and conditions can be confusing for the reader.

We agree with the reviewer that these figures are very dense in information. To help the reader, we have improved table 2 and included the name of each sample in that table.

We have done our best to make this figure comprehensible and improved the figure text to help the reader. We have made multiple versions both with and without color and text. Generally we do not prefer to give the same information in multiple ways (here color and text). However, when removing the text, it becomes very difficult to interpret the figure using only the colors. When we remove the colors, it becomes difficult to interpret which samples that originate from the same study participant.

We have improved the legend, but in the revised version we still have both text and color as we find this to be more clear and only 1 of 5 reviewers raised this point.

17. Figure 4: LUIN_28 is missing.

The colors can be a bit confusing, LUIN_28 is present (darker orange than LUIN_29). This is also why we do not want to remove the text labels. We have now added a clarification in table 2 of which subject had *in vivo*, *in vitro*, or both, samples included, which will hopefully make the interpretation of heatmaps and PCA plots easier.

18. Line 268: Figure 5 can be removed or moved to the Supplementary data. While it is useful to provide some background information, this information can easily be found elsewhere. Instead, it would be more informative to state that all transcripts were downregulated compared to specific reference genes.

Thank you for this comment. We have had the manuscript reviewed by 5 reviewers and as it was appreciated by a majority of the reviewers, we have kept it in the revised manuscript.

19. Figure 6: pfkA is down-regulated in vivo. The mRNA level is very low compared to other transcripts, and it is unclear whether this transcription is significant. The authors should provide information on the threshold (cut-off) used in their analyses? The same applies for napA, which has a p-value below 0.01.

See next reply.

20. Figure 6: Only 6 transcripts show a significant change when comparing in vitro and in vivo "expression" are compared. This finding contradicts Figure 5, which suggests that all gene products encoding enzymes shown in red and green colors are changed between in vitro and in vivo conditions. The authors should address this discrepancy and clarify the results.

Figure 6 has now been removed from the manuscript altogether, as well as the discussion of anaerobic vs aerobic which we had a hard time of fitting into the manuscript due to word limits (as the reviewer states in point 26, the discussion is quite long and had to be

shortened somehow). We will focus future work on subsets of the gene expression and this comment will be useful in that work.

21. Figure 7A: AspA is found to be significantly different between in vitro and in vivo conditions, but it is not shown in Figure 5. The authors should explain why AspA was not included in Figure 5 and provide a clear rationale for its inclusion in Figure 7A.

We based this figure on previous work by López-López et al. 2020 and by Othman et al. 2014. This is a simplified schematic of the TCA cycle, pentose-phosphate shunt and glycolysis of *H. influenzae*. As the reviewer correctly states, additional enzymes are interconnected with the ones shown in this figure, but all could not fit into this representation.

The original Figure 6 has now been removed and aspA is now not mentioned. This figure was mainly used as a comparison with previous works on the anaerobic and aerobic metabolism of *H. influenzae*, but since this discussion has now been removed, it is no longer needed.

22. Line 272: The authors state that "Iron transport and intracellular sequestering (2/11) were less common in vivo compared to in vitro (Figure 7B)." This statement is not accurate as the Authors only studied transcripts. Unless phenotypic data are available but not shown in the study, this statement appears to lack supporting evidence.

This sentence has been corrected.

23. Line 277: Similar to the comment above, the authors state that purine and pyrimidine metabolisms were changed. However, this statement is not accurate as the authors only analyzed transcripts. It would be more appropriate to state that transcripts involved in purine and pyrimidine metabolisms were changed.

This sentence has been corrected.

24. Line 286: The authors state that molybdate-transport was changed, but it is unclear whether this was tested. It would be helpful to clarify whether the statement is based on the transcript analysis alone or if there is additional evidence supporting this claim.

This sentence has been corrected.

25. Line 292: The authors state that many enzymes involved in a particular pathway were less expressed. It would be useful to indicate whether the authors checked the expression levels of these enzymes using western blots or enzymatic assays, and if not, to modify the statement to reflect that only transcript levels were analyzed.

Since expression can be used to also describe protein levels, we have now changed this sentence ("the gene expression of many enzymes.."). From the description of the study setup we hope that it is clear that no measurement of protein-levels or enzymatic assays

were performed. We also want to add that contrary to eucaryotes, prokaryotic gene expression is mainly regulated on the transcriptional level, even though post-transcriptional does sometimes occur.

26. Discussion is extensive long and needs to be shortened.

Thank you for this comment. We agree that the discussion is long and includes extensive elaboration about the data. The 5 reviewers have different opinions about the discussion (some wanted to add to it from the supplemental files or from other suggestions). We have now removed certain discussion parts altogether, for instance the sections about biofilm-formation, and aerobic vs anaerobic environment. It is now more concise, although still quite extensive.

27. Line 315-317: it is not surprising that bacteria are clustered based on growth conditions, despite diversity in genetic background, as core genes (housekeeping genes, Line 307) were targeted for DEG analysis.

We agree that this is not surprising. We also find some of the results to be intuitive, although it has not been shown for *H. influenzae* using the present experimental setup (transcriptome analysis of *in vitro* vs *in vivo* grown bacteria, many of the same genetic background). We hope that the data set will be useful for us and other researchers that want to make further analysis of the data and compare other growth conditions to the *in vivo* environment.

28. Line 332-333: the stage of diseases severity (status of inflammatory and immune response) might need to be considered here, as it may also affect the growth fitness or dormancy of bacteria during *in vivo*. Authors had mentioned the similar impact in Line 339-342.

Thanks for pointing this out. We've added that as a source of heterogeneity.

29. From my point of view, it is not strange that mRNA levels are different *in vivo* as compared to *in vitro* using excess NAD and hemin in the very rich BHI broth. The authors do not comment upon this.

We agree with this point. We've made the *in vitro* growth conditions more clear in the revised manuscript.

We would like to thank Reviewer 4 for the comments that were raised. Based on these points, we have improved the manuscript in the revised version and we think that this version more accurately describes the conditions in which the study was performed and that the results and discussion sections are of interest to the reader.

Reviewer #5 (Comments for the Author):

The manuscript Spectrum01639-23 by Polland et al presents an interesting approach to study the in vivo *H. influenzae* transcriptome in comparison to the in vitro obtained transcriptome, providing a good starting point for in-depth real-life approximation.

Because in vitro pure bacterial cultures are a more controlled and simple approach, researchers frequently overlook the fact that the culture conditions do not correspond to those encountered in patients. Although results from in vivo models are always more valuable, care must be taken when analyzing the data because this is a non-controlled environment with many external confusion factors that may affect the results.

The manuscript is clear and well-written. The authors clearly stated their objectives and used the appropriate methodology to achieve the results. The manuscript is well-structured, and the results are clear. The supplementary data adds value to this work by reporting and discussing in detail some additional results that did not fit into the main manuscript document. The work has been well planned, with an emphasis on avoiding or reducing external and methodological factors, however, I have some concerns regarding the use of the *H. influenzae* clinical strains.

My first doubt regards the indistinctive use of non-typeable *H. influenzae* (NTHi) and capsulated strains; there are clear differences between both types of strains, but NTHi are the most commonly identified in respiratory infections. Besides, only some serotypes were included (B, D and F). I did not see in the description of the nine studied strains whether they are all NTHi or if there are any capsulated strains (this information, along with MLST information, should be included in Table 2). If all of the test strains are NTHi, consider making the core and pan genome using only NTHi strains.

Thank you for raising this point. The original typing analyses were made based on transcriptome assemblies. We have now sequenced all cultured strains and the MLST based on this new data has been added to table 2.

Regarding the choice of strains to include in the generation of pan and core genomes, we have included a section in the discussion about this choice and expanded both the methods and results section describing the creation of the reference genomes. We reasoned that the 15 strains listed in the KEGG GENOME database would be a good balance and give a fairly reduced core genome, while the pan genome would still be limited. Adding more strains would expand the pan genome for each added strain, but the core genome did not change much when adding additional strains. Since we focused on basic biological function of *H. influenzae*, the use of the core genome created from different strains was of higher interest than the capsule type.

Using an external set of input genomes has the strength that we can reuse the same core genome in follow up studies. However, we look forward to do pair-wise comparisons using the entire genome of each included strain. We have added a discussion about this in the revised manuscript.

Second, after reading the entire manuscript I have some doubts about the origin of the *H. influenzae* used in vitro. Theoretically, the strains used for in vitro testing should come from the samples studied in vivo; however, the methodology explaining the origin of these strains is ambiguous, and the results (Figure 2, color legends) show that there is only coincidence in five of the nine strains (LUIN_26; LUIN_33; MAAK_03; MAIV_24; MAIV_32). On the other hand, LUIN_28 was only studied in vitro, while LUIN_28, LUIN_29, LUIN_31 and MAIV_34 were only studied in vivo. Despite that, in lines 219 to 221 say "... the in vitro cultured bacterial isolates cluster closely together regardless of genetic background. The in vivo gene expression showed a higher transcriptional diversity and did not cluster with the corresponding bacterial isolated cultured in vitro". This sentence implies that gene expression was studied in vivo and in vitro on the same strains. This should be revised because, in order to draw any conclusions, the strains used in vitro should come from their original clinical samples in order to reduce differences. It is obvious that clinical samples may contain more than one *H. influenzae* lineage (this cannot be controlled experimentally) but the ones isolated and used for experimental testing should be directly linked to the in vivo testing.

Thank you for this point about the samples included in the study. We have clarified how samples were selected in the main text, as well as in the revised Fig 1 and table 2. We have also modified the discussion about this.

All clinical samples were sent to the Clinical microbiology laboratory for bacterial cultures. A protocol was set up with the lab and identified *H. influenzae* were to be isolated and stored. This was frequently overlooked by the staff, and some isolates were thus not saved.

However, we believe the consequences of this for the current study are small, since we have limited the main differential gene expression analysis to the core genome that is shared between the different strains.

Table 2 shows the presence of two different *H. influenzae* lineages (ST-12 and a new MLST) in sample LUIN_26. Which strain was used in the in vitro tests? Or were they both used? The ST numbers of the newly identified MLSTs should be included (ask for the ST number on the MLST page).

The MLST scheme has now been updated using newly performed DNA-sequencing data (instead of assembled and aligned RNA transcripts). The original MLST were done on transcript assemblies. The two lab cultured LUIN_26 had the same MLST in the new analysis based on genome assemblies.

However, it is interesting that the RNA-profiles differ between the two LUIN_26 samples. We don't know the reason for this and even though they had the same MLST, they could represent two lineages. Other explanations could be that there might have been a pipetting error or that these two did somehow experience small variations in growth conditions and therefore differed in their gene expression from one another.

Minor comments

Lines 201-202: "... including genomes from NTHi serotype A, B, D and F..." In Supplementary Table S1, describing the strains used for Core and Pan genome determination, there is no serotype A strain.

Thank you for noting this, it has been corrected. Also, Rd KW20 was initially serotype d, but over the years in different laboratories, it has lost its capsule and is now NTHi. This is corrected in the revised table and text.

Lines 201-202: "... including genomes from NTHi serotype A, B, D and F..." This sentence brings confusion because it seems that NTHi have serotypes, use instead "... including genomes from NTHi and serotypes b, d and f".

This has been changed in the revised version of the text. Less focus is now on serotypes as this study is not focused on the differences between serotypes.

Once again, we would like to express our gratitude for your time and expertise in reviewing our manuscript. We believe that the revisions we have implemented address all of the concerns raised during the review process and have significantly strengthened the overall quality of the study. We are confident that our findings will make a valuable contribution to the field.

We look forward to hearing your feedback on the revised manuscript.

July 12, 2023

Dr. Magnus Paulsson
Lunds Universitet
Infection medicine, Department of Clinical Sciences Lund, Medical faculty
Lund
Sweden

Re: Spectrum01639-23R1 (In vivo gene expression profile of *Haemophilus influenzae* during human pneumonia)

Dear Dr. Magnus Paulsson:

Your manuscript has been accepted, and I am forwarding it to the ASM Journals Department for publication. You will be notified when your proofs are ready to be viewed.

In the meantime, please ensure that you have made the RNA Seq dataset publicly available in the ENA if this has not already been done.

Sincerely,

John Attack
Editor, Microbiology Spectrum
